# Mechanisms of iron- and O₂-sensing by the [4Fe-4S] cluster of the global iron regulator RirA

**Ma Teresa Pellicer Martinez[1†], Jason C Crack[1†], Melissa YY Stewart[1†], Justin M Bradley[1], Dimitri A Svistunenko[2], Andrew WB Johnston[3], Myles R Cheesman[1], Jonathan D Todd[3], Nick E Le Brun[1]\***

[1]Centre for Molecular and Structural Biochemistry, School of Chemistry, University of East Anglia, Norwich, United Kingdom; [2]School of Biological Sciences, University of Essex, Colchester, United Kingdom; [3]School of Biological Sciences, University of East Anglia, Norwich, United Kingdom

**Abstract** RirA is a global regulator of iron homeostasis in *Rhizobium* and related α-proteobacteria. In its [4Fe-4S] cluster-bound form it represses iron uptake by binding to IRO Box sequences upstream of RirA-regulated genes. Under low iron and/or aerobic conditions, [4Fe-4S] RirA undergoes cluster conversion/degradation to apo-RirA, which can no longer bind IRO Box sequences. Here, we apply time-resolved mass spectrometry and electron paramagnetic resonance spectroscopy to determine how the RirA cluster senses iron and $O_2$. The data indicate that the key iron-sensing step is the $O_2$-independent, reversible dissociation of $Fe^{2+}$ from $[4Fe-4S]^{2+}$ to form $[3Fe-4S]^0$. The dissociation constant for this process was determined as $K_d = \sim3~\mu M$, which is consistent with the sensing of 'free' iron in the cytoplasm. $O_2$-sensing occurs through enhanced cluster degradation under aerobic conditions, via $O_2$-mediated oxidation of the $[3Fe-4S]^0$ intermediate to form $[3Fe-4S]^{1+}$. This work provides a detailed mechanistic/functional view of an iron-responsive regulator.
DOI: https://doi.org/10.7554/eLife.47804.001

**\*For correspondence:**
n.le-brun@uea.ac.uk

[†]These authors contributed equally to this work

**Competing interests:** The authors declare that no competing interests exist.

## Introduction

Iron–sulphur clusters are ubiquitous protein cofactors that play essential roles across all of life in processes as diverse as respiration, photosynthesis, and DNA replication (*Beinert et al., 1997*; *Johnson et al., 2005*), with recent evidence that they may be even more abundant than initially thought (*Rouault, 2015*). Elucidating the precise nature of their roles is a major challenge, which is often complicated by the extreme reactivity of the cluster to $O_2$ and other gases. However, this very sensitivity has been exploited through the evolution of iron–sulphur cluster-containing transcriptional regulators that enable cells to sense and respond to, for example, oxidative stress and changes in concentrations of metabolically important species such as $O_2$ and iron (*Beinert and Kiley, 1999*; *Crack et al., 2014a*).

Iron is an essential micronutrient for nearly all of life, but is also potentially extremely toxic due to its ability to catalyse, via redox cycling, the formation of damaging reactive oxygen species. Consequently, for life to flourish, not only must sufficient iron be obtained from the environment, but its precise form must be carefully controlled (*Andrews et al., 2003*). A central part of this control is exerted through the regulation of iron uptake into the cell in response to intracellular iron levels. In many bacteria, including such taxonomically diverse model organisms as *Escherichia coli* and *Bacillus subtilis*, iron uptake is under the control of the global iron regulator Fur (Ferric uptake regulator). Iron is sensed through the availability and binding of $Fe^{2+}$ directly to the Fur protein, resulting in a

**eLife digest** Virtually all life forms require iron to survive, yet too much of the metal can be catastrophic. In healthy cells, many systems regulate this delicate balance. For instance, when a protein called RirA is active in *Rhizobium* bacteria, it can sense high levels of the metal and helps to shut down the production of proteins that bring in more iron. RirA contains a cluster of four iron and four sulphur atoms, which acts as a sensor for iron availability: however, exactly how this cluster structure detects the levels of the metal in a cell was previously unclear.

Pellicer Martinez, Crack, Stewart et al. used a technique known as time-resolved mass spectrometry to examine the sensory response of the iron-sulphur cluster of RirA when different levels of iron were available. The results revealed a 'loose' iron atom in the cluster; when iron levels fall down, this atom is rapidly lost as it is scavenged for use in other essential processes. Without it, the cluster in RirA collapses and the protein becomes inactive. This prompts the cell to produce proteins that enable it to take up iron from its surroundings. Once iron levels are high, RirA can regain its cluster and is active again, stopping the production of proteins that bring in more iron.

Iron-sulphur clusters are common in many proteins, and this work offers new insight into their various roles. It also highlights the potential to use time-resolved mass spectrometry to examine biological processes in depth.

DOI: https://doi.org/10.7554/eLife.47804.002

conformation that can bind *cis*-acting 'Fur boxes' close to the promoters of genes encoding proteins that function in the iron-uptake machinery, causing (usually) repression of transcription (*Lee and Helmann, 2007*; *Pohl et al., 2003*). The direct binding of $Fe^{2+}$ as a co-repressor also occurs in DtxR (Diphtheria toxin Repressor) from *Corynebacterium diphtheriae* and related species. Although DtxR has no significant sequence similarity to Fur, it shares significant structural features, in addition to $Fe^{2+}$-binding (*D'Aquino et al., 2005*; *Ding et al., 1996*).

*Rhizobium* and other closely related rhizobial genera that induce nitrogen-fixing nodules on legumes, as well as the pathogens *Bartonella*, *Brucella*, and *Agrobacterium*, contain a very different global iron regulator called RirA (Rhizobial iron regulator A) (*Todd et al., 2002*; *Wexler et al., 2003*; *Yeoman et al., 2004*; *Todd et al., 2005*; *Todd et al., 2006*; *Rudolph et al., 2006*; *Chao et al., 2005*; *Ngok-Ngam et al., 2009*; *Viguier et al., 2005*; *Bhubhanil et al., 2014*; *Crespo-Rivas et al., 2019*), which has no structural or sequence similarity whatsoever to Fur or DtxR. Rather, RirA belongs to the Rrf2 super-family of transcriptional regulators (*Keon et al., 1997*) that includes IscR (regulator of iron–sulphur cluster biosynthesis) (*Rajagopalan et al., 2013*; *Schwartz et al., 2001*) and NsrR (regulator of nitrosative stress response) (*Crack et al., 2012*; *Crack et al., 2015*; *Volbeda et al., 2017*), both of which are homodimeric and bind an iron–sulphur cluster. In *Rhizobium leguminosarum*, the symbiont of peas, beans and clovers, RirA regulates many genes involved in iron homeostasis, by binding to *cis*-acting 'IRO boxes' (*Yeoman et al., 2004*). Recently, it was shown that *R. leguminosarum* RirA binds a [4Fe-4S] cluster and that this form of the protein binds to the IRO box sequence (*Pellicer Martinez et al., 2017*). Although the structure of RirA is not yet available, a model based on the structure of dimeric [4Fe-4S] NsrR, the first reported for any cluster-bound Rrf2 regulator (*Volbeda et al., 2017*), is shown in *Figure 1*. Exposure of [4Fe-4S] RirA to low iron conditions and/or $O_2$ resulted in cluster conversion to generate a [2Fe-2S] form, which binds to the IRO box with lower affinity than the [4Fe-4S] form. The [2Fe-2S] form was also unstable under low iron conditions, resulting in apo-RirA, which did not bind the IRO box sequence (*Pellicer Martinez et al., 2017*).

To understand how RirA functions as an iron sensor and how it enables the cell to integrate iron and $O_2$ signals requires that the mechanisms by which it responds to these two signals are elucidated. Traditional approaches to studies of iron–sulphur cluster proteins cannot readily resolve intermediates of cluster conversion/decay. ESI-MS utilizing solution and ionisation conditions under which proteins remain folded enables accurate mass detection of intact proteins and protein complexes, and has been used extensively to study protein-protein interactions, interactions of proteins with drugs, nucleic acids, sugars and lipids, as well as protein structural changes (*Liko et al., 2016*; *Hopper and Robinson, 2014*; *Rodenburg et al., 2017*). It has also been used to study protein-

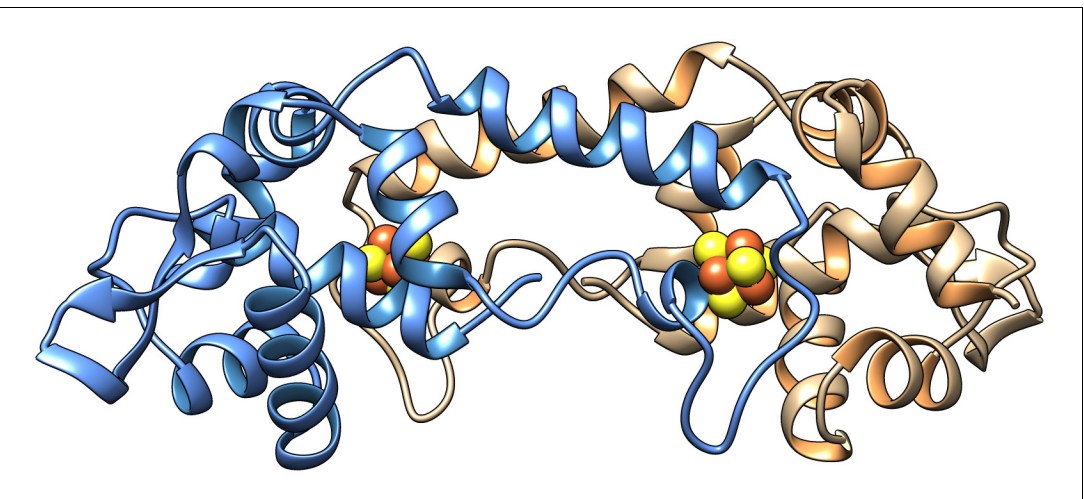

**Figure 1.** Model of the homodimer form of [4Fe-4S] RirA. Cartoon ribbon representation showing one protomer in blue and the other in beige, with iron and sulphide ions in brown and yellow, respectively. The model was generated using Swiss-Model (*Bienert et al., 2017*; *Pettersen et al., 2004*) based on the structure of *Streptomyces coelicolor* [4Fe-4S] NsrR (PDB: 5N07) (*Volbeda et al., 2017*). An amino acid residue alignment between RirA and NsrR is shown in *Figure 1—figure supplement 1*.

DOI: https://doi.org/10.7554/eLife.47804.003

The following figure supplement is available for figure 1:

**Figure supplement 1.** Amino acid residue alignment of *R. leguminosarum* RirA and *S. coelicolor* NsrR.

DOI: https://doi.org/10.7554/eLife.47804.004

cofactor interactions where the cofactor remains bound following ionisation, and amongst these, ESI-MS of metalloregulators (*Glauninger et al., 2018*) and iron–sulphur cluster proteins (*Johnson et al., 2000*) and their reactivity (*Crack and Le Brun, 2019*) have also been shown to be valuable. Time-resolved mass spectrometry has been exploited for studying protein metal ion binding (*Ngu and Stillman, 2006*) and heme transfer (*Tiedemann et al., 2012*), and for structural studies using H/D exchange (*Konermann et al., 2011*). Recently, time-resolved ESI-MS provided detailed mechanistic information of the $O_2$-sensing reaction of the [4Fe-4S] cluster binding FNR regulator (*Crack et al., 2017*).

Here, we employed time-resolved ESI-MS under non-denaturing conditions, along with EPR spectroscopy, to elucidate the exact series of molecular events involved in the conversion of the RirA cluster in response to key environmental signals. The data reveal, in remarkable detail, the nature of intermediates formed and differences in cluster conversion under aerobic and anaerobic conditions. Importantly, we find that the key sensing step, the *initiation* of cluster conversion/degradation, is dependent on $Fe^{2+}$ but is independent of $O_2$, and that the $K_d$ for the fourth iron binding to the cluster is in the low micromolar range, supporting a primary role for RirA as an iron sensor in vivo. Nevertheless, $O_2$ influences the overall rate of cluster decay through the oxidation of a key cluster intermediate and of cluster-derived sulphide. Together, the data provide a comprehensive mechanistic and functional view of an important iron-responsive regulator.

## Results and discussion

### Mass spectrometric analysis of [4Fe-4S] RirA cluster conversion under anaerobic conditions

Regular electrospray ionisation mass spectrometry (ESI-MS) in water/acetonitrile acidified with formic acid results in denatured proteins and the loss of non-covalently attached cofactors. However, the use of a volatile buffer at physiological pH is known to preserve the protein's folded state and, with it, any bound cofactors (*Crack et al., 2017*). Therefore, using ESI-MS under such non-denaturing

conditions, we set out to observe in real time the decay of the [4Fe-4S] RirA cluster in response to low iron conditions and to determine the identity of any intermediary cluster conversion breakdown products. The feasibility of this approach was indicated by the previously reported ESI-MS of [4Fe-4S] RirA (*Pellicer Martinez et al., 2017*), in which both dimeric and monomeric forms were observed. Ionisation of RirA in the MS experiment results in partial dissociation of the RirA dimer into monomers. Initially, conditions for the formation of the monomeric form were optimised because this enabled the unambiguous assignment of cluster species. Analysis of the RirA dimer follows in a later section.

The initial spectrum, *Figure 2A* (black line), in which the protein was held under non-denaturing anaerobic, iron-replete conditions, was very similar to that previously described for cluster-bound RirA (*Pellicer Martinez et al., 2017*); the major peak at 17,792 Da corresponds to the [4Fe-4S] form (see *Supplementary file 1* for predicted and observed masses), but a range of lower intensity cluster breakdown species was also present, which are most likely due to cluster damage sustained during the exchange of the protein into the volatile buffer necessary for ESI-MS studies. Indeed, these were observed to decay away while the [4Fe-4S] form remained stable (*Figure 2—figure supplement 1*), consistent with previous studies that demonstrated that the [4Fe-4S] form of RirA is entirely stable under anaerobic, iron-replete conditions (*Pellicer Martinez et al., 2017*).

We previously showed that various $Fe^{2+}$ chelators, including EDTA and Chelex 100, can simulate low iron conditions by efficiently competing for $Fe^{2+}$. The lack of dependence on the concentration or type of chelator used indicated that this competition occurs through two equilibria corresponding to the loss of iron from the cluster and subsequent binding by the chelator. Consistent with this, under low iron conditions generated by the anaerobic addition of 250 µM EDTA, major changes in the ESI-MS spectrum were observed after 30 min, *Figure 2A*. ESI-MS changes were then measured over this time period (t = 0 is the spectrum prior to the addition of chelator, t = 30 is 30 min post addition). While the [4Fe-4S] peak was observed to decay away, peaks corresponding to protein-bound cluster fragments (17,586–17,762 Da) initially increased in intensity. These included [4Fe-3S], [3Fe-4S], [3Fe-3S], [3Fe-2S], [3Fe-S], [2Fe-2S] and [2Fe-S] forms (see *Figure 2B* and *Supplementary file 1*). These species then subsequently decayed away (*Figure 2C*), with a peak at 17442 Da, due to apo-RirA, becoming the major feature of the spectrum. Peaks at +16, 32 and 64 Da, due to oxygen or sulphur adducts of apo-RirA, were also observed, though at relatively low intensity. The full time course is shown as a 3D plot in *Figure 2D*.

To assist with unambiguous assignment of cluster-bound forms of RirA, [4Fe-4S] RirA with cluster sulphide specifically labelled with [34]S was generated via in vitro cluster synthesis (cluster reconstitution) using [34]S-L-cysteine (*Crack et al., 2019*). The major peak in the deconvoluted ESI-MS spectrum was at 17,800 Da, shifted +8 Da relative to the natural abundance (predominantly [32]S) form of [4Fe-4S] RirA, *Figure 3A* and *Supplementary file 1*. Exposure of the [34]S form to EDTA under anaerobic conditions resulted in a set of peaks, due to intermediate species, observed at masses shifted relative to those of the [32]S sample, *Figure 3B–G*. The peaks initially assigned to [3Fe-4S], [4Fe-3S], [3Fe-3S], [3Fe-2S], [3Fe-S], [2Fe-2S] and [2Fe-S] species were shifted by +2 Da for each sulphide, confirming the identification of these intermediate/product forms. As expected, the apo-protein peak was not shifted (*Figure 3H*).

The apo-RirA adduct at +16 and +32 Da were also not shifted, suggesting that these are due, respectively, to one and two O adducts, while the peak at +64 Da was shifted by + 4 Da, consistent with a double sulphur adduct derived from cluster sulphide (*Figure 3—figure supplement 1*). The latter arises as a result of oxidation of cluster sulphide, generating $S^0$, which can be incorporated into Cys thiol side chains. The origin of oxygen adducts is less clear. Recent ESI-MS studies of the anaerobic nitrosylation of NsrR also revealed O adduct species (*Crack and Le Brun, 2019*), and we note that under certain conditions aqueous samples may generate oxygen in the ESI source (*Banerjee and Mazumdar, 2012*).

It is well established that non-denaturing ESI-MS can be used to follow chemical processes in solution (e.g. ligand binding to a protein or isotope exchange over time), yielding quantitative thermodynamic and kinetic information (*Hopper and Robinson, 2014*; *Ngu and Stillman, 2006*; *Crack et al., 2017*; *Pacholarz et al., 2012*; *Schermann et al., 2005*). To determine the sequence of events in RirA [4Fe-4S] cluster degradation, abundances of the different cluster fragment species in *Figure 3* were analysed as a function of time. *Figure 4A* shows plots of relative intensities due to

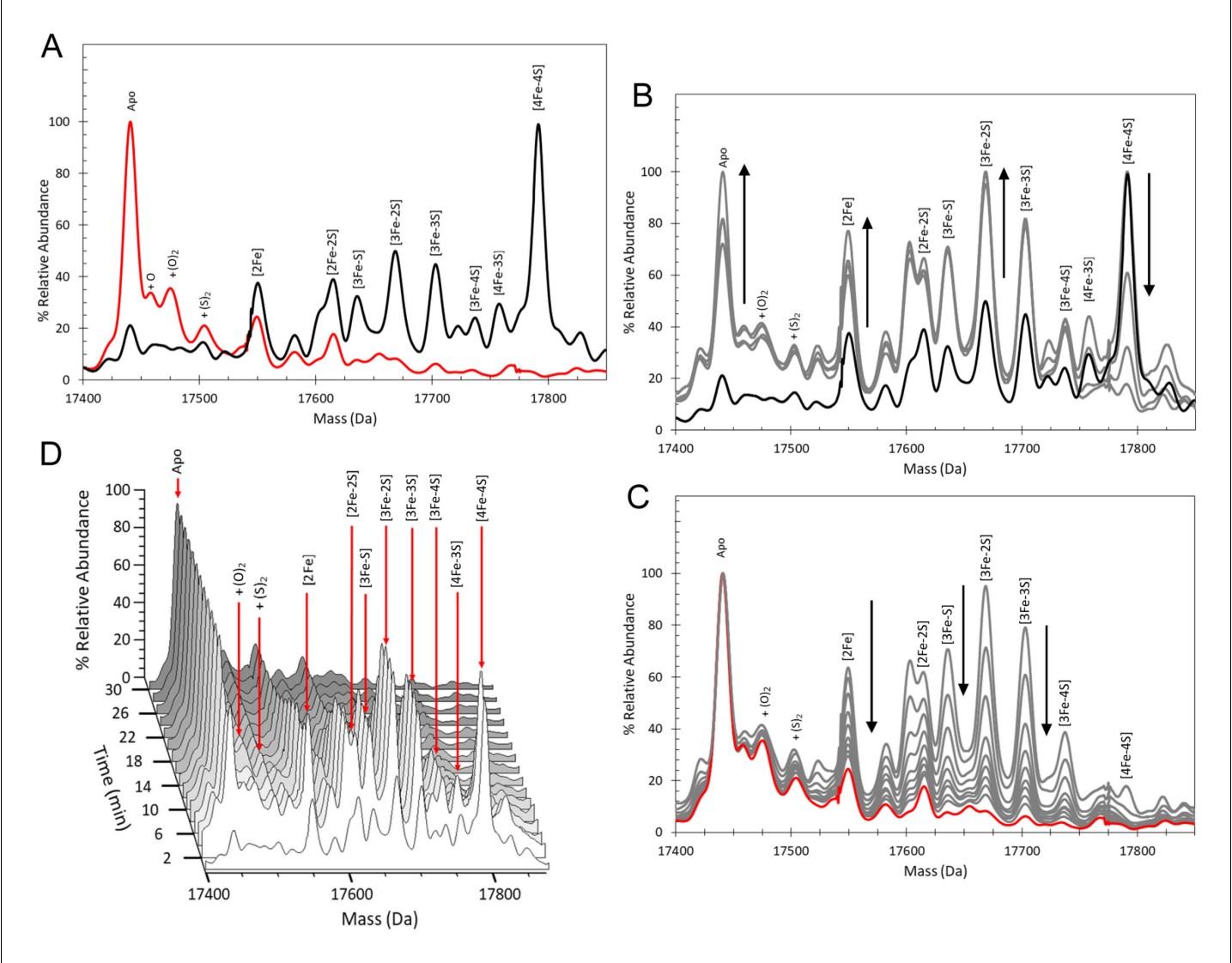

**Figure 2.** ESI-MS analysis of [4Fe-4S] RirA cluster conversion under anaerobic, low iron conditions. (A) Deconvoluted (to neutral mass) mass spectrum of [4Fe-4S] RirA prior to the addition of 250 μM EDTA (t = 0, black line) and 30 min after the addition at 37°C (t = 30, red line). (B) and (C) Deconvoluted mass spectra measured at intervening times: 0–8 min (from the point of EDTA addition, spectra recorded at 2 min intervals) (B) and 8–30 min (C). Starting and endpoint spectra are in black and red, respectively (corresponding to the data in (A). (D) 3D plot of time dependent changes in the ESI-MS spectrum showing the formation and decay of RirA cluster intermediates and formation of apo-RirA products. [4Fe-4S] RirA (~25 μM) was in anaerobic 250 mM ammonium acetate pH 7.3. Low iron conditions were generated by the addition of 250 μM EDTA and cluster conversion/degradation was followed at 37°C.

DOI: https://doi.org/10.7554/eLife.47804.005

The following figure supplement is available for figure 2:

**Figure supplement 1.** Analysis of RirA [4Fe-4S] cluster breakdown species in the absence of an $Fe^{2+}$ chelator.

DOI: https://doi.org/10.7554/eLife.47804.006

[4Fe-4S] and [3Fe-4S] species as a function of time following the addition of EDTA, while *Figure 4B–E* show equivalent plots for [3Fe-3S], [3Fe-2S], [2Fe-2S] and apo-RirA species.

The data show that [4Fe-3S] and [3Fe-4S] were the first intermediates to reach their maximum abundance, consistent with their formation early in the conversion process. The [3Fe-3S] species was the next intermediate to maximise, followed by [3Fe-2S] species. The last of the intermediates to maximise in abundance and the last to decay away was [2Fe-2S] RirA. This was previously shown to be a stable intermediate following the [4Fe-4S] to [2Fe-2S] conversion promoted by EDTA

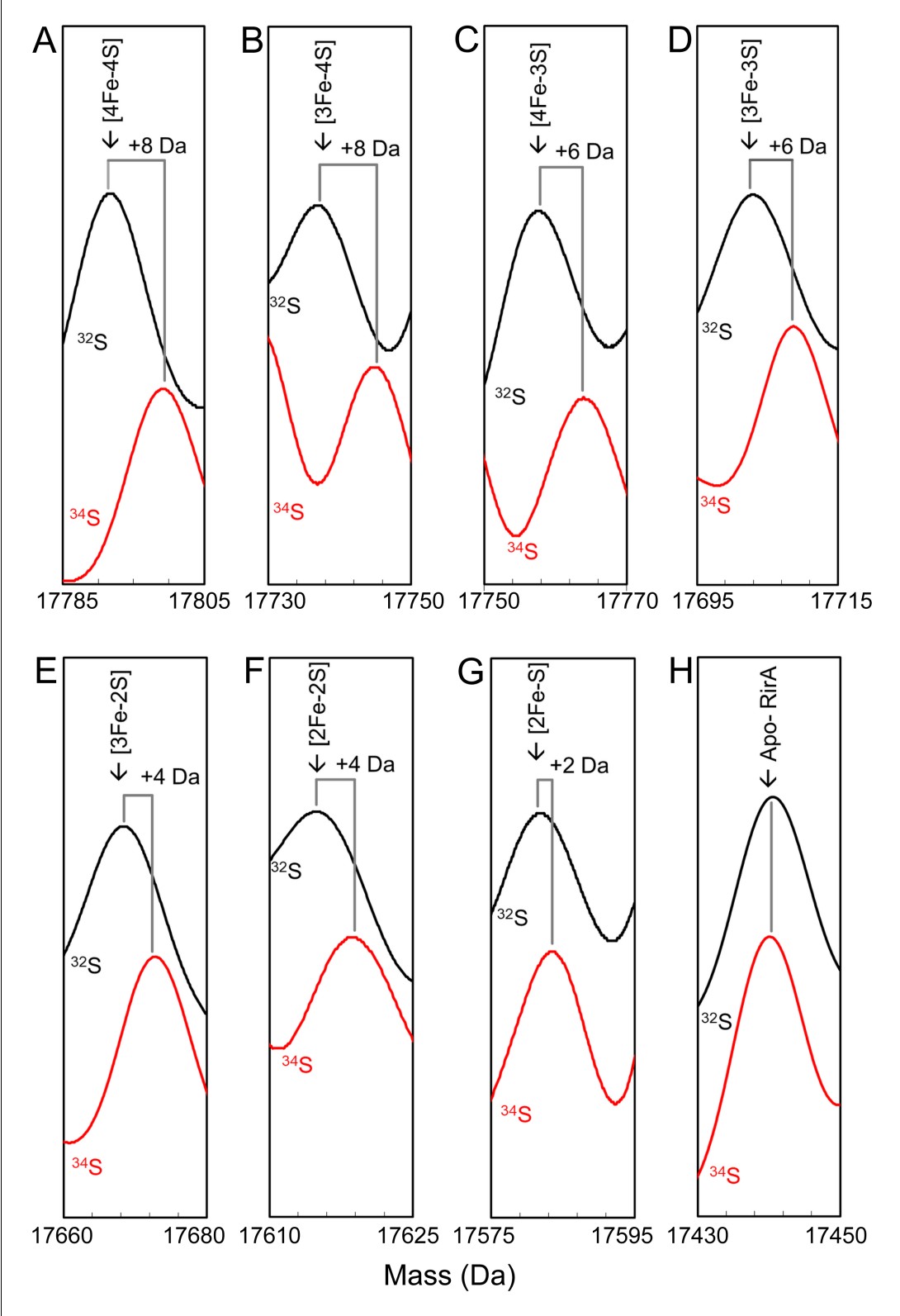

**Figure 3.** Mass shifts observed for the RirA [4Fe-4S] cluster, conversion intermediates and cluster products upon $^{34}$S substitution of cluster sulphides. (A–G) Deconvoluted mass spectra of natural abundance sulphur [4Fe-4S] RirA and cluster conversion intermediates (black lines) and the equivalent $^{34}$S-substituted forms (red lines), as indicated. (H) As in A) except spectra show peaks due to apo-RirA resulting from natural abundance and $^{34}$S [4Fe-4S]

*Figure 3 continued on next page*

*Figure 3 continued*

RirA samples, as indicated. Predicted mass shifts for the assigned species are indicated. Note that the ESI-MS data for natural abundance and $^{34}$S-substituted [4Fe-4S] RirA were previously published (*Crack et al., 2019*); see also *Supplementary file 1*.

DOI: https://doi.org/10.7554/eLife.47804.007

The following figure supplement is available for figure 3:

**Figure supplement 1.** Assignment of apo-RirA adduct species using isotope substitution following degradation of the [4Fe-4S] cluster under low iron/anaerobic conditions.

DOI: https://doi.org/10.7554/eLife.47804.008

(*Pellicer Martinez et al., 2017*). However, the buffer conditions used in the present study, and which are required for the ESI-MS experiment, appear to reduce the stability of the [2Fe-2S] form leading to observation of apo-protein (*Figure 4E*), as recently observed following conversion of the O$_2$ sensor [4Fe-4S] FNR to its [2Fe-2S] form (*Crack et al., 2017*).

The observed sequence of cluster intermediates provided an outline for the mechanism of cluster conversion/degradation, which was used as the basis for global analysis of multiple (n = 4) mass spectrometric kinetic data sets. Iterative optimisation of the mechanism/global fit resulted in the reaction scheme shown in *Figure 5*, with corresponding fits of the peak intensities due to the formation and/or decay of [4Fe-4S], [4Fe-3S], [3Fe-4S], [3Fe-3S], [3Fe-2S] and [2Fe-2S] RirA species (see the solid lines in the plots of *Figure 4*). Rate constants obtained from the fit to the reaction scheme are given in *Table 1*.

The mechanistic scheme indicates that loss of a single iron or sulphide ion appears to be an obligatory first step in the cluster conversion process. However, the rate constant describing the loss of a sulphide to generate [4Fe-3S] is extremely low, indicating that this reaction does not occur as a significant part of the (chemical) conversion mechanism and that the observation of a [4Fe-3S] species (which is present at time zero following addition of EDTA) results from damage to the cluster during ionisation. It is important to note that the extent of cluster damage is low and this does not affect the mechanistic picture of the cluster conversion/degradation reaction that emerges. Thus, loss of an initial iron, [4Fe-4S] to [3Fe-4S], is the heavily favoured route for the initiation of cluster conversion. Another important feature of the global fit is that the first step, loss of Fe$^{2+}$ from [4Fe-4S]$^{2+}$, is reversible.

The overall rate of cluster conversion observed here is significantly higher than that previously reported from absorbance kinetic experiments (*Pellicer Martinez et al., 2017*). To confirm that ESI-MS and absorbance spectroscopy report on the same process, cluster degradation was followed by monitoring A$_{382\ nm}$ (corresponding to the maximum absorbance of the [4Fe-4S] cluster) as a function of time for a [4Fe-4S] RirA sample in volatile ammonium acetate buffer (necessary for ESI-MS measurements) rather than the HEPES buffer previously used (*Pellicer Martinez et al., 2017*), see *Figure 4F*. The decay fitted well to a bi-exponential function, where the initial, more rapid phase must correspond to the loss of the [4Fe-4S] cluster (in which it converts to a species that cannot be identified by its absorbance alone), followed by a slower phase that corresponds to further decay of the cluster. The rate constant for the initial phase, $k = 0.34$ min$^{-1}$, is in excellent agreement with the rate constant ($k = 0.30$ min$^{-1}$) for the conversion of [4Fe-4S] to [3Fe-4S], derived from the ESI-MS kinetic data (*Table 1*). Though [4Fe-4S] and [3Fe-4S] clusters often have similar absorbance extinction coefficients, loss of Fe$^{2+}$ from the RirA [4Fe-4S]$^{2+}$ cluster would be expected to result in some change in the UV-visible absorbance spectrum, as observed here and for a similar process in aconitase (*Emptage et al., 1983*). Thus, the enhanced rate of conversion reported here is not a consequence of the ESI-MS method, but rather reflects the different buffer conditions used here compared to those of the previous study (*Pellicer Martinez et al., 2017*). The RirA cluster is clearly more labile in the ammonium acetate buffer, most likely because acetate is a weak iron chelator.

The ESI-MS data show that the [3Fe-3S] RirA species was formed from [3Fe-4S] and [4Fe-3S] clusters with similar observed rate constants ($k = 0.090$ and $0.087$ min$^{-1}$, respectively). The temporal behaviour of the [3Fe-3S] intermediate, where it was observed to maximise at ~4 min before decaying away, is consistent with it being an intermediate in the [4Fe-4S] to [2Fe-2S] cluster conversion pathway. A novel [3Fe-3S] intermediate was recently observed by mass spectrometry during the O$_2$-

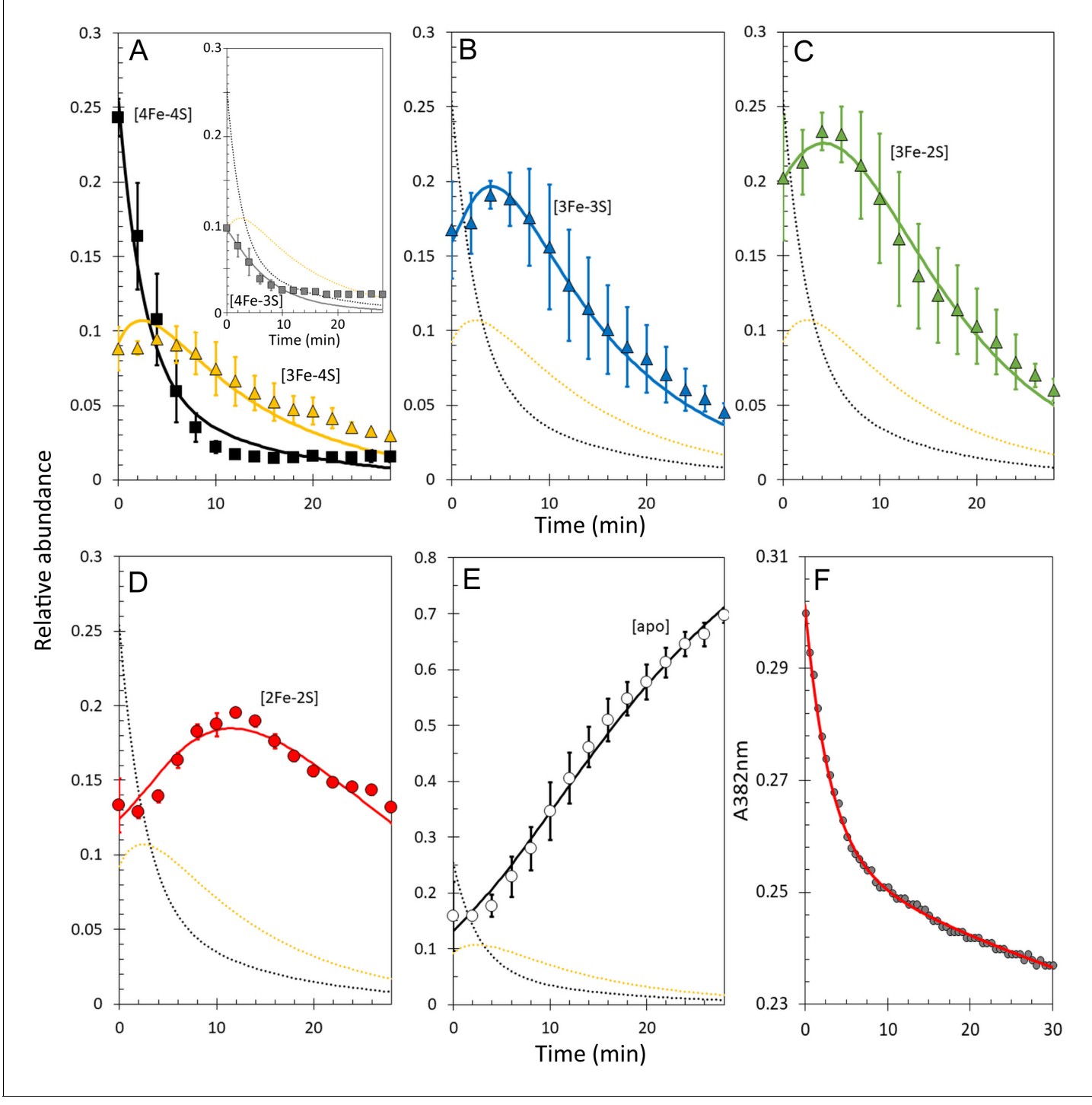

**Figure 4.** Kinetic analysis of [4Fe-4S] RirA cluster conversion/degradation. (**A**) Plots of relative abundances of [4Fe-4S] cluster (black) and [3Fe-4S] (yellow) species as a function of time following exposure to 250 μM EDTA at 37° C. Inset is a plot of relative abundances of the [4Fe-3S] cluster, illustrating that it is likely formed during ionisation. (**B**) – (**E**) As in (**A**) but showing [3Fe-3S] (**B**), [3Fe-2S] (**C**), [2Fe-2S] (**D**) and apo- (**E**) forms of RirA. Fits of the data, generated by a global analysis of the experimental data based on the reaction scheme depicted in *Figure 5*, are shown as solid lines. Broken lines correspond to the kinetic profile of the cluster species associated with that colour and are included to permit easy comparison between intermediates. Error bars show standard error for ESI-MS datasets (n = 4, derived from one biological replicate and three technical replicates). (**F**) Plot of $A_{382\ nm}$ versus time following addition of 250 μM EDTA to [4Fe-4S] RirA (30 μM in cluster in 250 mM ammonium acetate, 500 μM glutathione, pH 7.3) at 37° C. The red line indicates a fit of the data generated using a bi-exponential function. We note that significant $A_{382\ nm}$ remains after 30 min, where

*Figure 4 continued on next page*

*Figure 4 continued*

ESI-MS indicates that the majority of the protein is in an apo-form. The residual absorbance most likely arises from Fe/S species present in the cuvette, either attached to the protein, or in solution/suspension, for example as iron sulphide or iron acetate (*Pellicer Martinez et al., 2017*).

DOI: https://doi.org/10.7554/eLife.47804.009

dependent [4Fe-4S] to [2Fe-2S] cluster conversion reaction of FNR (*Crack et al., 2017*). The observation of a similar intermediate here suggests that it may be a more widespread feature of [4Fe-4S] cluster conversions. While the structure of this particular [3Fe-3S] intermediate was not established, that of a small molecule $[3Fe-3S]^{3+}$ cluster was recently published, revealing a planar hexagonal arrangement of alternating iron and sulphide ions (*Lee et al., 2016*). If the [3Fe-3S] intermediate of RirA has a similar planar structure (*Figure 5*), it would reveal a key feature of the [4Fe-4S] to [2Fe-2S] cluster conversion, in which the geometry of the coordinating ligands changes from tetrahedral (for the [4Fe-4S] cluster) to planar (for the [2Fe-2S] cluster).

[3Fe-3S] RirA degraded predominantly to form [2Fe-2S] ($k = 0.5$ min$^{-1}$), before finally generating apo-RirA ($k = 0.07$ min$^{-1}$). The mechanistic scheme indicates that [3Fe-3S] may also decay to [3Fe-2S] and then on to apo-RirA (in part via [2Fe-2S]). It is recognised that additional intermediate species are likely to be involved in the degradation of [3Fe-2S]/[2Fe-2S] to apo-RirA. While a [2Fe-S] form was observed in mass spectra, its temporal behaviour could not be sensibly modelled, suggesting that it might, at least in part, arise from spontaneous re-assembly of cluster fragments as iron/sulphide ions are released during cluster conversion/degradation.

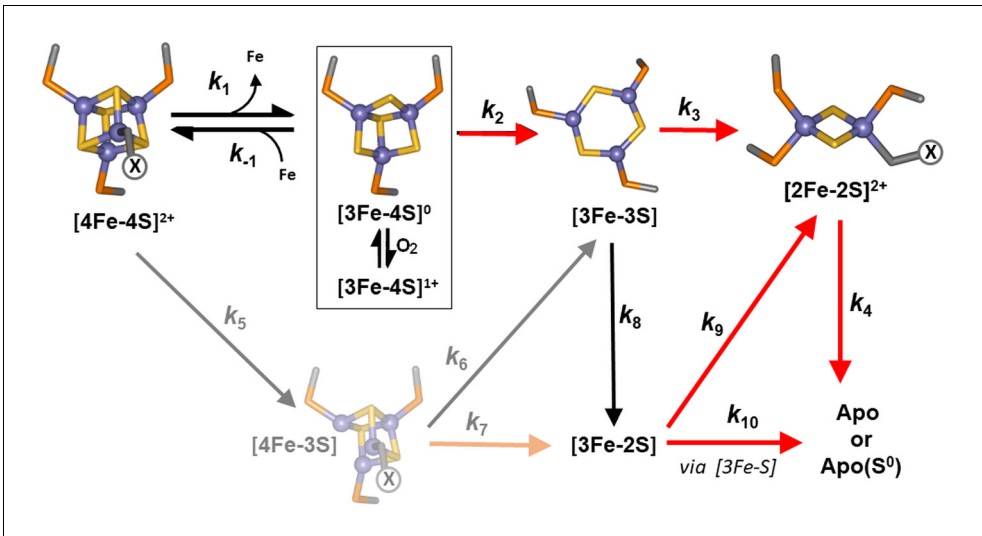

**Figure 5.** Proposed reaction scheme for [4Fe-4S] RirA cluster conversion/degradation. Reaction scheme used to fit time-resolved ESI-MS data for [4Fe-4S] RirA. Values of rate constants ($k$) are given in *Table 1*. Note that the ESI-MS data cannot distinguish between $[3Fe-4S]^{1+}$ and $[3Fe-4S]^{0}$ forms (boxed), so these are treated only as [3Fe-4S] in the kinetic model. They can be distinguished by EPR, however (see below), and the kinetic EPR data support the proposal that $[3Fe-4S]^{0}$ is susceptible to oxidation to $[3Fe-4S]^{1+}$ in the presence of $O_2$. Reactions that are enhanced in rate (i.e. have increased associated rate constants) in the presence of $O_2$ are indicated by red arrows. The initial [4Fe-4S] cluster is coordinated by three Cys residues and one unknown ligand, illustrated in the figure as 'X'. The structure of the [3Fe-3S] species proposed here is based on the recently reported small molecule [3Fe-3S] species in which iron and sulphur form a hexagonal ring (*Lee et al., 2016*). The faded portion of the figure (involving steps marked $k_5$, $k_6$ and $k_7$) represents a minor pathway that most likely arises from ionisation-induced damage.

DOI: https://doi.org/10.7554/eLife.47804.010

**Table 1.** Rate constants resulting from global fit of experimental ESI-MS data using the model shown in **Figure 5**.

| Reaction step | Rate constant (min$^{-1}$)$^{*,†}$ | | Reaction step |
| --- | --- | --- | --- |
| | Anaerobic (-$O_2$) | Aerobic (+$O_2$) | |
| $k_1$ | 0.300 ± 0.010 | 0.320 ± 0.020 | [4Fe-4S] → [3Fe-4S] + Fe |
| $k_{-1}$ | 4.67 ± 0.33 × $10^3$ | 4.67 ± 0.43 × $10^3$ | [3Fe-4S] + Fe → [4Fe-4S] |
| $k_2$ | 0.090 ± 0.002 | 0.230 ± 0.010 | [3Fe-4S] → [3Fe-3S] |
| $k_3$ | 0.500 ± 0.010 | 1.200 ± 0.050 | [3Fe-3S] → [2Fe-2S] |
| $k_4$ | 0.070 ± 0.001 | 0.200 ± 0.005 | [2Fe-2S] → apo |
| $k_5$ | 0.008 ± 0.002 | 0.007 ± 0.002 | [4Fe-4S] → [4Fe-3S] |
| $k_6$ | 0.087 ± 0.003 | 0.087 ± 0.008 | [4Fe-3S] → [3Fe-3S] |
| $k_7$ | 0.083 ± 0.001 | 0.150 ± 0.008 | [4Fe-3S] → [3Fe-2S] |
| $k_8$ | 0.030 ± 0.001 | 0.026 ± 0.020 | [3Fe-3S] → [3Fe-2S] |
| $k_9$ | 0.044 ± 0.004 | 0.140 ± 0.002 | [3Fe-2S] → [2Fe-2S] |
| $k_{10}$ | 0.160 ± 0.004 | 0.300 ± 0.010 | [3Fe-2S] → apo |

$^{*}$With the exception of $k_{-1}$, which is a second order rate constant with units of $M^{-1}$ $min^{-1}$.
$^{†}$Standard errors are indicated.
DOI: https://doi.org/10.7554/eLife.47804.011

## The effect of $O_2$ on [4Fe-4S] RirA cluster conversion revealed by mass spectrometry

Previous studies of [4Fe-4S] RirA showed that the cluster is sensitive to $O_2$, undergoing conversion in an apparently similar manner to that observed under low iron conditions (**Pellicer Martinez et al., 2017**). To gain a much more detailed view of this aspect of the conversion process, [4Fe-4S] RirA was investigated using non-denaturing ESI-MS under low iron conditions in the presence of $O_2$ (228 µM). Cluster conversion/breakdown species were observed to form and decay as a function of time, **Figure 6** and **Table 1**. Plots of the individual cluster intermediate species as a function of time, **Figure 7**, revealed marked differences to those in the absence of $O_2$. We note that the presence of O adducts in experiments carried out under anaerobic conditions suggested that some $O_2$ might be generated during the ESI-MS experiment; however, the differences observed between the anaerobic and aerobic experiments suggest that this is relatively minor. Indeed, the formation of several intermediates, including the [3Fe-3S] and [2Fe-2S] forms, occurred significantly more rapidly under aerobic conditions, consistent with previous observations that a combination of low iron and $O_2$ enhanced the rate of cluster conversion/degradation (**Pellicer Martinez et al., 2017**). Interestingly, the formation of sulphur adducts of apo-RirA, which were confirmed to be derived from cluster sulphide by $^{34}$S-dependent mass shifts (**Figure 6—figure supplement 1**), occurred to a much greater extent in the presence of $O_2$, consistent with $O_2$ acting as oxidant for $S^{2-}$ ions released from the cluster (**Figure 6**). Low abundance peaks due to oxygen adducts (the first at apo-RirA +16 Da) were also observed (**Figure 6—figure supplement 1**). The temporal behaviour of the double sulphur adduct of apo-RirA is shown in **Figure 7E**.

A global analysis of the aerobic kinetic ESI-MS data was achieved on the basis of the same mechanistic scheme employed for the analysis of anaerobic data (**Figure 5**) - see the solid lines in the kinetic plots of **Figure 7**. The rate constants obtained from the fit are consistent with the enhanced rates of several steps in the conversion mechanism in the presence of $O_2$ (**Table 1**). As a control, absorbance spectroscopy was again used to monitor the decay of $A_{382\ nm}$ as a function of time for an aerobic [4Fe-4S] RirA sample in ammonium acetate buffer (**Figure 7F**). In contrast to the equivalent anaerobic experiment (**Figure 4F**), the data were fitted well by a single exponential function with a rate constant, $k$ = 0.34 min$^{-1}$, again in excellent agreement with that obtained for the initial, [4Fe-4S] to [3Fe-4S] step of the cluster conversion reaction (**Table 1**).

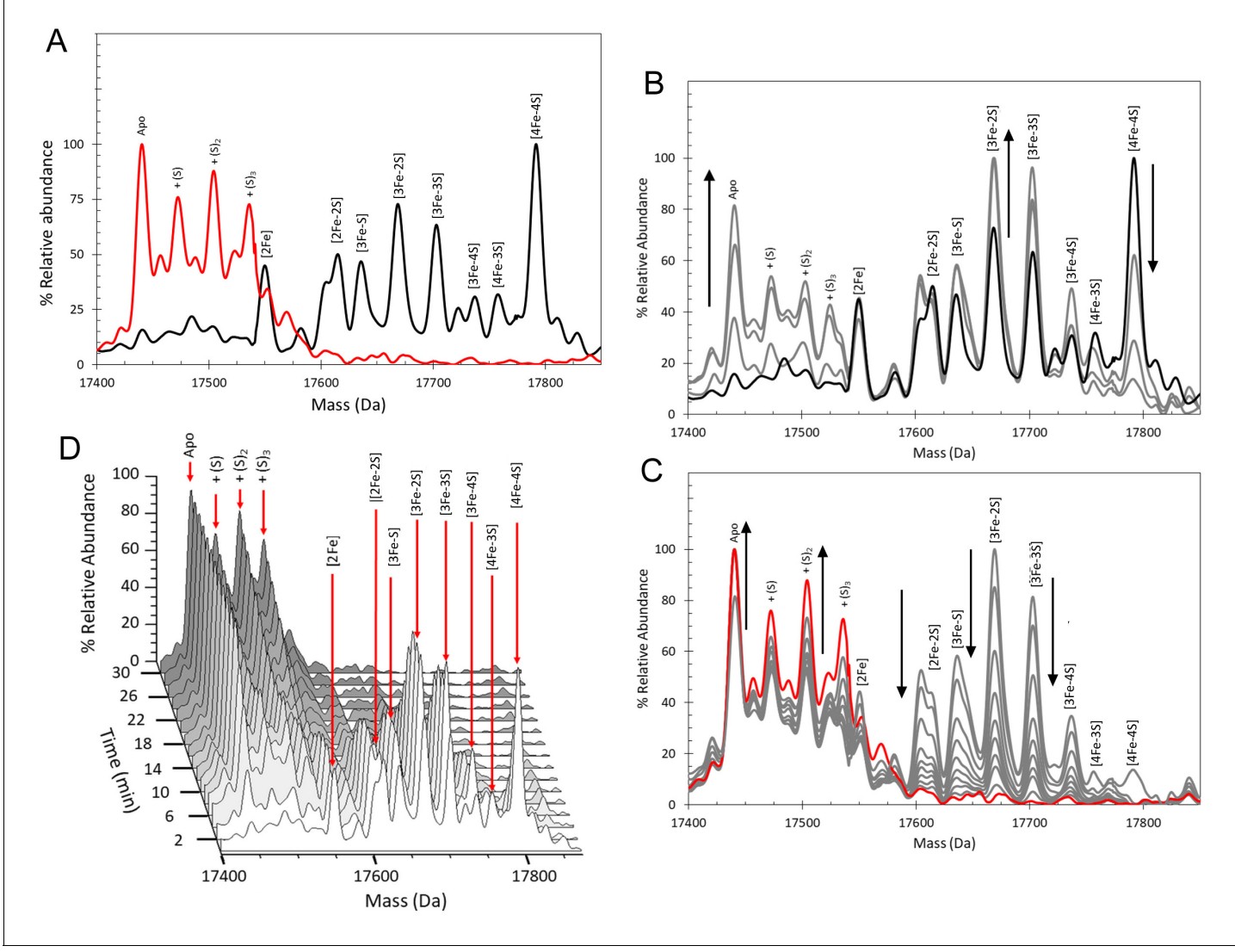

**Figure 6.** ESI-MS analysis of [4Fe-4S] RirA cluster conversion under aerobic, low iron conditions. (A) Deconvoluted mass spectrum of [4Fe-4S] RirA prior to the addition of 250 μM EDTA (black line) and 30 min after the addition at 37°C (red line). (B) and (C) Deconvoluted mass spectra measured at intervening times: 0–6 min (relative to the addition of EDTA/$O_2$, spectra recorded at 2 min intervals) (B) and 6–30 min (C). Starting and endpoint spectra are in black and red, respectively (corresponding to the data in (A). (D) 3D plot of time-dependent changes in the ESI-MS spectrum showing the formation and decay of RirA cluster intermediates and formation of apo-RirA products. [4Fe-4S] RirA (~25 μM) was in aerobic (~228 μM $O_2$) 250 mM ammonium acetate pH 7.3. Low iron conditions were generated by the addition of 250 μM EDTA and cluster conversion/degradation was followed at 37°C.

DOI: https://doi.org/10.7554/eLife.47804.012

The following figure supplement is available for figure 6:

**Figure supplement 1.** Assignment of apo-RirA adduct species using isotope substitution following degradation of the [4Fe-4S] cluster under low iron/aerobic conditions.

DOI: https://doi.org/10.7554/eLife.47804.013

## The effect of $O_2$ on [4Fe-4S] RirA cluster conversion revealed by EPR spectroscopy

Previous studies of RirA revealed that a paramagnetic species was formed in RirA samples under oxidative conditions (*Pellicer Martinez et al., 2017*). Thus, EPR spectroscopy was employed here to monitor the formation of paramagnetic intermediates as a function of time following exposure to

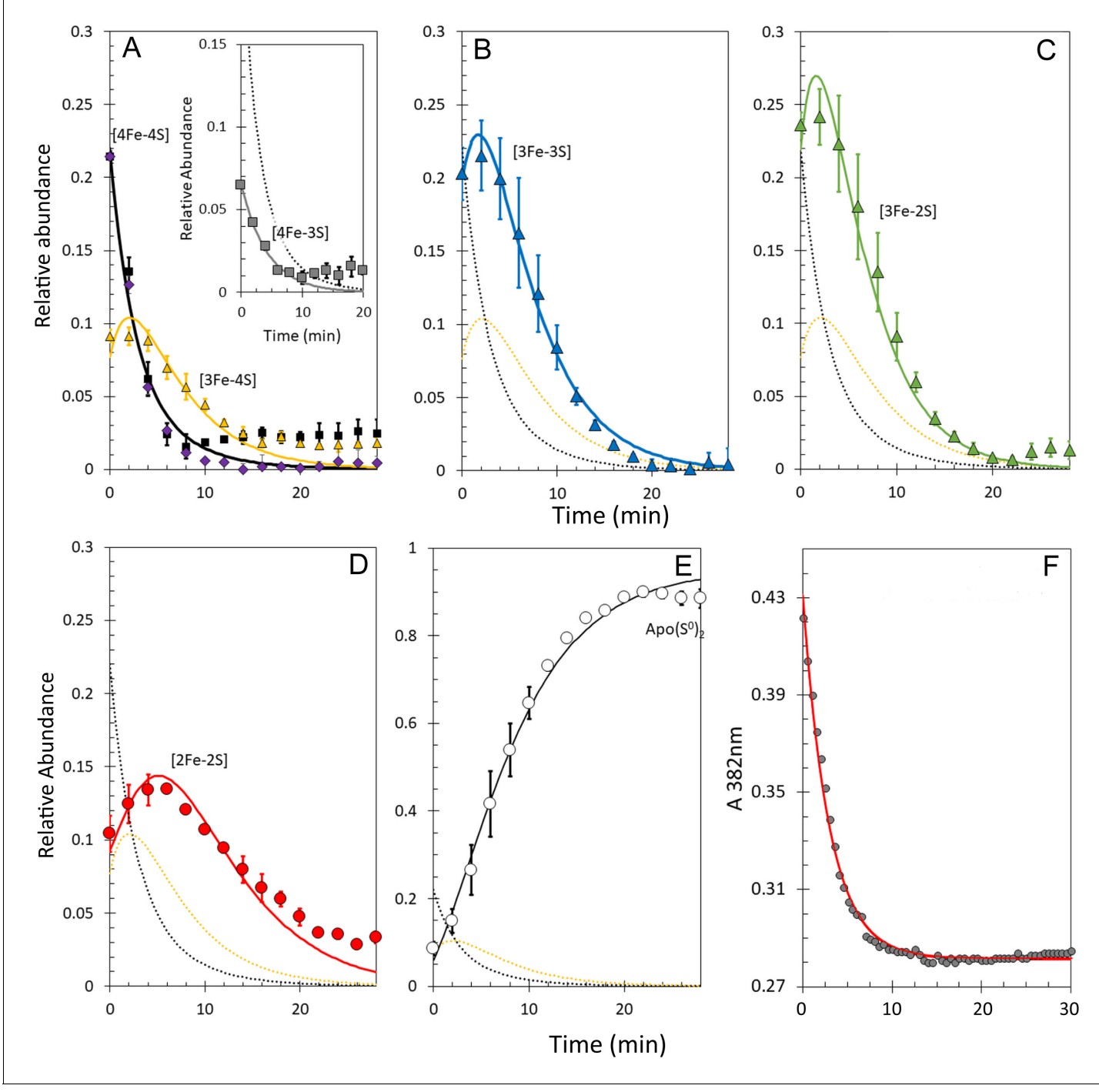

**Figure 7.** Kinetic analysis of [4Fe-4S] RirA cluster conversion/degradation in the presence of $O_2$. (A) Plots of relative ESI-MS abundances of [4Fe-4S] cluster (black) and [3Fe-4S] (yellow) species as a function of time following exposure to 250 µM EDTA and 228 µM $O_2$ at 37° C, as shown in *Figure 6*. Normalised $A_{382\ nm}$ data (see panel F) are plotted in purple, revealing the close correspondence between the absorbance and [4Fe-4S] decay. Inset is a plot of ESI-MS data for the [4Fe-3S] species. (B) – (D) As in (A) but showing plots of relative abundances of [3Fe-3S] (B), [3Fe-2S] (C), and [2Fe-2S] (D) cluster intermediates. (E) Plot of relative abundance of the double sulphur adduct of apo-RirA. Fits of the data generated by a global analysis of the ESI-MS data, based on the reaction scheme depicted in *Figure 5*, are shown as solid lines. Broken lines correspond to the kinetic profile of the cluster species associated with that colour and are included to permit easy comparison between intermediates. Error bars show standard error for ESI-MS datasets (n = 2 derived from two technical replicates). (F) Plot of $A_{382\ nm}$ versus time following addition of 250 µM EDTA and 228 µM $O_2$ to [4Fe-4S] RirA (30 µM in cluster) at 37° C. The red line indicates a fit of the data generated using a single exponential function. The origin of the residual $A_{382\ nm}$ at 30 min, discussed in the legend of *Figure 4*, may be somewhat different in the presence of $O_2$, where significantly more sulphide undergoes oxidation.
DOI: https://doi.org/10.7554/eLife.47804.014

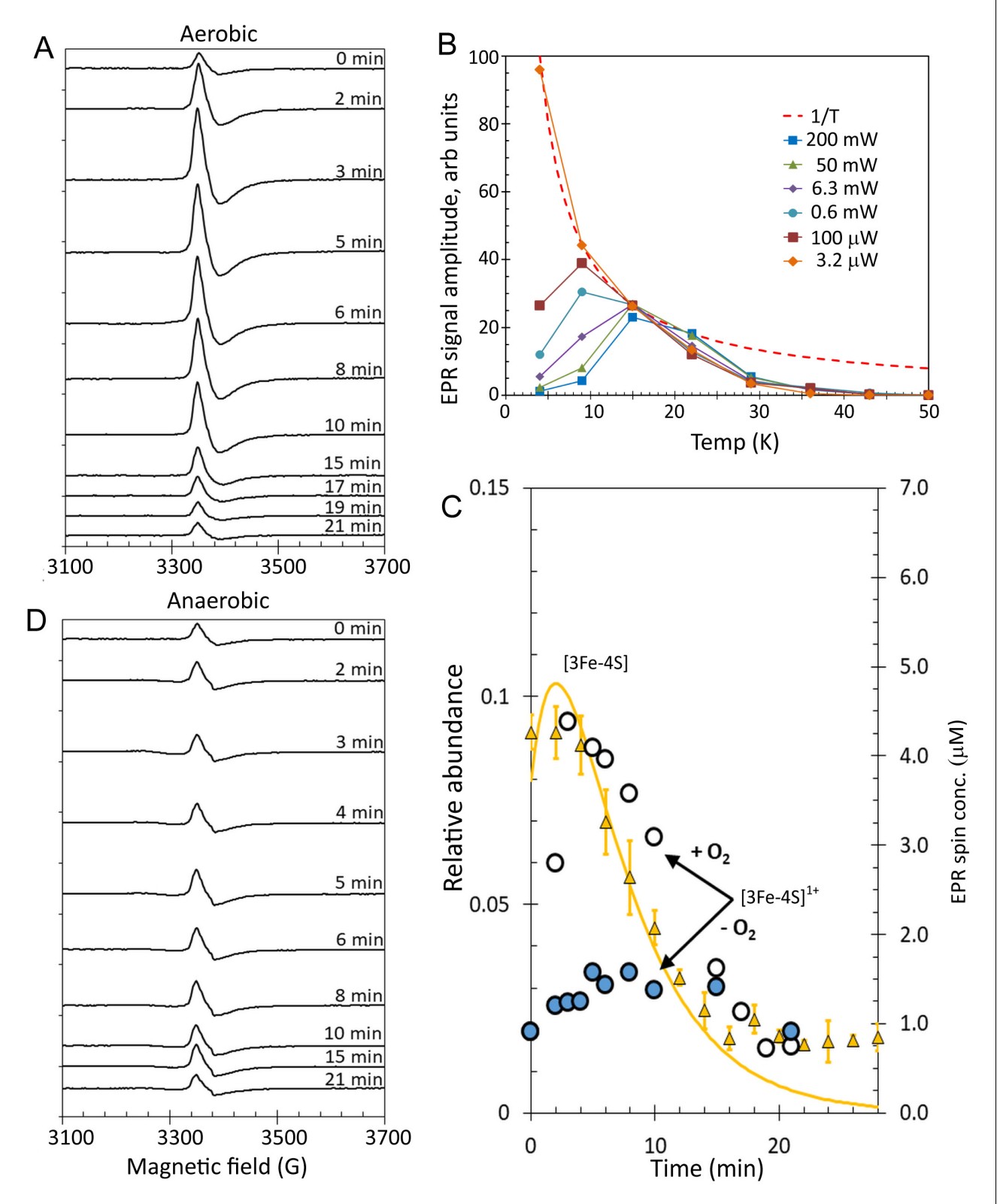

**Figure 8.** EPR analysis of [4Fe-4S] RirA cluster conversion under aerobic and anaerobic low iron conditions. (**A**) EPR spectra of [4Fe-4S] RirA at specified time points following the addition of 250 μM EDTA under aerobic conditions at 37°C. (**B**) Plot of g = 2.01 EPR signal intensity dependence on temperature at a range of microwave powers. The dashed red line represents a hyperbolic 1/T function plotted in arbitrary units to represent the Curie law (when no saturation effects take place). All experimental dependences are individually scaled so that each meets the hyperbola via at least one data

*Figure 8 continued on next page*

*Figure 8 continued*

point, with remaining data points lying below the hyperbola (indicating that the signal saturates). (C) Plot of ESI-MS relative abundances of RirA [3Fe-4S] cluster as a function of time following exposure of RirA [4Fe-4S] to 250 μM EDTA at 37° C and aerobic conditions (yellow triangles) and RirA [3Fe-4S]$^{1+}$ cluster determined under identical condition by EPR under aerobic (+$O_2$, white-filled circles) and anaerobic (-$O_2$, blue-filled circles) conditions. The solid yellow line represents the global fit to the ESI-MS data (as shown in *Figure 7*). (D) As in (A) but under anaerobic conditions. [4Fe-4S] RirA (~25 μM) was in aerobic (~228 μM $O_2$) 250 mM ammonium acetate pH 7.3.

DOI: https://doi.org/10.7554/eLife.47804.015

The following figure supplements are available for figure 8:

**Figure supplement 1.** EPR saturation characteristics of the RirA EPR signal.
DOI: https://doi.org/10.7554/eLife.47804.016

**Figure supplement 2.** Temperature dependences of the RirA EPR signal at different microwave powers.
DOI: https://doi.org/10.7554/eLife.47804.017

low iron conditions in the presence of $O_2$. To facilitate direct comparison of kinetic data, samples were prepared at the same concentration, in the same buffer and at the same temperature as used for the mass spectrometric studies. An S = ½ signal at g = 2.01, similar to that previously reported for RirA (*Pellicer Martinez et al., 2017*), was observed, which was assigned to the [3Fe-4S]$^{1+}$ cluster intermediate (*Figure 8A*). However, the detection of a [3Fe-3S] intermediate also raised the possibility that this species might be responsible for the EPR signal, as recently proposed for cluster conversion in [4Fe-4S] FNR (*Crack et al., 2017*).

So, to investigate the EPR-active species in more detail, the saturation properties of the g = 2.01 EPR signal were studied as a function of microwave power and temperature (*Figure 8—figure supplement 1*). Signal intensities as a function of temperature at six microwave power values were plotted (*Figure 8B* and *Figure 8—figure supplement 2*). The inclusion of a 1/T hyperbola (*Figure 8B*) reveals the narrow temperature interval in which the EPR signal follows the Curie law (based on non-saturated and non-broadened lines originating from an undisturbed Boltzmann distribution of energy levels populations) for each microwave power. At a very low microwave power, such as 3.2 μW, the EPR signal perfectly obeyed the Curie law in the explored temperature range of 4–15 K. As the power was increased, the temperature dependence in the 4–9 K region became lower, indicating power saturation at these temperatures for all powers ≥ 100 μW. At temperatures above ~20 K, all dependences deviated from the Curie hyperbola, which is an effect of line broadening. No microwave power was able to produce any detectable EPR absorbance at temperatures above 50 K. This is a particular characteristic of [3Fe-4S]$^{1+}$ clusters; they exhibit the most sensitive dependence on increasing temperature, with signal disappearing at a lower temperature than for any other cluster type (*Svistunenko et al., 2006*). Such microwave power saturation behaviour, as well as the g-value and EPR line shape are highly characteristic of well-characterised [3Fe-4S]$^{1+}$ clusters (*Beinert and Thomson, 1983*; *Cammack, 1992*) and so we conclude that, in the case of RirA, the EPR active species is the [3Fe-4S]$^{1+}$ cluster. It is perhaps surprising that none of the other intermediates identified by ESI-MS are detected by EPR; it appears that such species are diamagnetic.

To determine the kinetic properties of the RirA [3Fe-4S]$^{1+}$ intermediate, EPR measurements were performed for samples prepared at a range of time points following exposure to aerobic low iron conditions, *Figure 8A*. Quantification of the g = 2.01 signal revealed that ~ 1 μM (~4%) of the cluster was present in the [3Fe-4S]$^{1+}$ form at time zero, prior to the introduction of chelator and $O_2$. The signal intensity increased with time up to ~4.5 μM (~18% of the original cluster concentration) by 3 min before decaying away over the next 15 min (*Figure 8A and C*). The [3Fe-4S]$^{1+}$ cluster contains three $Fe^{3+}$ ions, meaning that it is formed from a [4Fe-4S]$^{2+}$ cluster through loss of a $Fe^{2+}$ ion and oxidation of the remaining $Fe^{2+}$ to $Fe^{3+}$. EPR measurements were also carried out for [4Fe-4S] RirA samples under identical conditions except that $O_2$ was omitted (*Figure 8D*). A signal similar to that observed in the aerobic experiment was detected, but at a significantly lower concentration. Beginning at ~4% of original cluster concentration, the signal increased only to ~6.5% after 5 min, then remained at that level up to ~15 min before returning to its pre-EDTA exposure concentration (*Figure 8C and D*). We note that the [3Fe-4S]$^{1+}$ signal does not entirely decay away under either

aerobic or anaerobic conditions. One possibility is that this low intensity residual signal represents an 'off pathway' form of the cluster that is less reactive.

## The first step of 4Fe-4S] RirA cluster conversion is the $O_2$-independent reversible loss of $Fe^{2+}$ to form [3Fe-4S]

While the presence of $O_2$ enhances several of the steps of cluster conversion/degradation, importantly, this is not the case for the initial *reversible* step in which the cluster loses a $Fe^{2+}$ ion to form the [3Fe-4S] intermediate: rate constants for the initial step (principally loss of iron) are similar under aerobic and anaerobic conditions. However, the oxidation state of the [3Fe-4S] cluster is affected by the presence of $O_2$. A comparison of the ESI-MS data for [3Fe-4S] from *Figure 7B* and the EPR intensity data for [3Fe-4S]$^{1+}$ is shown in *Figure 8C*. The data are generally in good agreement for the aerobic experiment. There is a slight lag in the appearance of the EPR signal relative to the ESI-MS [3Fe-4S] peak, which could arise from the initial formation of the EPR silent [3Fe-4S]$^0$ intermediate, followed by its oxidation to the [3Fe-4S]$^{1+}$ form in the presence of $O_2$. In the anaerobic experiment, the observed intensity of the [3Fe-4S]$^{1+}$ cluster is much lower, consistent with the major form being the EPR silent [3Fe-4S]$^0$ form. In combination, the ESI-MS and EPR data show that a [3Fe-4S] cluster intermediate is formed in both the absence and presence of $O_2$ (at similar abundance as judged by ESI-MS), but it is predominantly present in different oxidation states, being largely in the reduced [3Fe-4S]$^0$ form under anaerobic conditions and in the oxidised [3Fe-4S]$^{1+}$ form under aerobic conditions.

The insensitivity of the initial step to $O_2$ revealed here by time-resolved ESI-MS is consistent with the previous observation that the rate of the [4Fe-4S] to [2Fe-2S] cluster conversion reaction is $O_2$-independent, leading to the conclusion that the rate-limiting step of the reaction does not involve $O_2$ (*Pellicer Martinez et al., 2017*). While the overall rates of reaction in the ammonium acetate buffer used here for ESI-MS and EPR experiments are greater than those previously reported, the *global* analyses indicate that, when $O_2$ is present, the initial reaction (corresponding to the loss of a single iron (as $Fe^{2+}$) from the [4Fe-4S]$^{2+}$ cluster) is, or is close to, the rate-limiting step (*Table 1*). This accounts for why the absorbance decay at 382 nm follows a single exponential in the presence of $O_2$ but a bi-exponential in the absence of $O_2$: in the former condition, the initial and subsequent reaction ([3Fe-4S] to [3Fe-3S]) occur at similar rates, while under anaerobic conditions, the second step is significantly slower. The independence from $O_2$ of the initial step is consistent with RirA functioning principally as an iron sensor.

In considering the question of why $O_2$ should significantly affect the [3Fe-4S] to [3Fe-3S] conversion, it is likely that the [3Fe-4S]$^{1+}$ species resulting from $O_2$-mediated oxidation of the initially formed [3Fe-4S]$^0$ cluster is more labile, and differences in oxidation state of intermediate cluster irons could be an important stability factor throughout. Another factor that is likely to be of at least equal importance is the enhanced oxidation of cluster sulphide in the presence of $O_2$, which would be expected to accelerate the degradation of the [3Fe-4S], [3Fe-3S] and [2Fe-2S] forms towards apo-RirA. A comparison of the rate constants in *Table 1* suggests that both sulphide and iron oxidation processes are important for the overall enhanced rate of degradation towards apo-RirA in the presence of $O_2$.

## Determination of the $K_d$ for binding of the fourth iron of the RirA [4Fe-4S] cluster

The data presented above point to the equilibrium [4Fe-4S]$^{2+}$ ↔ [3Fe-4S]$^0$ + $Fe^{2+}$, as being key for iron sensing. If this is the case, then the binding affinity of the fourth iron should reflect the likely 'free' iron concentration of the cell, to which it would need to respond. Levels of 'free' or chelatable iron have not been reported for *R. leguminosarum*, but have been measured for *Escherichia coli*, in which it was found to be ~10 μM (*Keyer and Imlay, 1996*), and there are no a priori reasons why those in other proteobacteria should be very different.

Mag-fura-2, which binds $Fe^{2+}$ to form a 1:1 complex with a $K_d$ of 2.05 μM, has been used previously to probe $Fe^{2+}$-binding to proteins (*Rodrigues et al., 2015*). A titration of [4Fe-4S] RirA with an increasing concentration of mag-fura-2 was carried out. Importantly, the concentration of $Fe^{2+}$-mag-fura-2 formed at the end of the titration corresponded to ~1 iron per [4Fe-4S] cluster (*Figure 9*), confirming that the process being measured corresponded to the [4Fe-4S]$^{2+}$ ↔ [3Fe-4S]$^0$ + $Fe^{2+}$

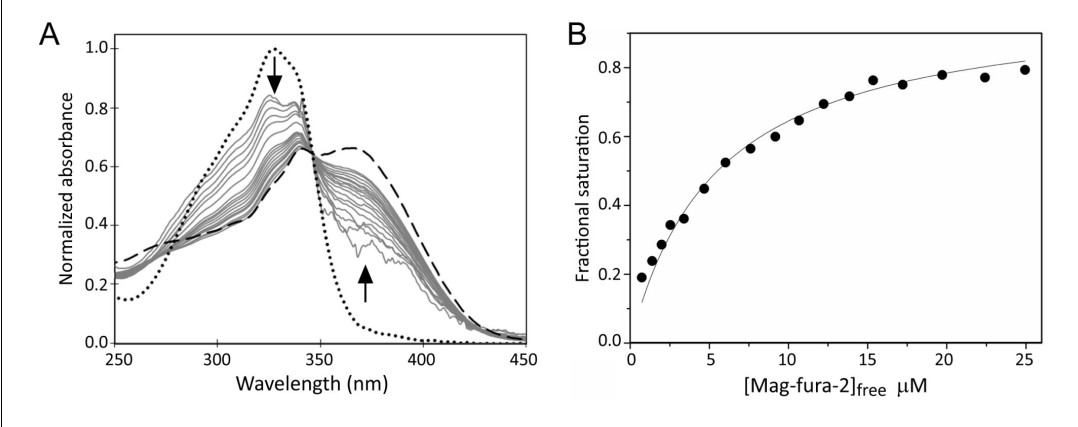

**Figure 9.** Determination of the binding affinity of the fourth iron of the RirA [4Fe-4S] cluster. (**A**) Normalised UV-visible absorbance spectra showing the ratio of apo- and $Fe^{2+}$-bound mag-fura-2 upon addition of increasing concentrations of mag-fura-2 to [4Fe-4S] RirA (7 μM) was in 25 mM HEPES, 50 mM NaCl, 750 mM KCl, pH 7.5. In addition to the titration spectra (solid grey lines), spectra of fully apo- (dashed line) and $Fe^{2+}$-bound (dotted line) forms of mag-fura-2 are shown. Arrows denote the direction of absorbance change upon increasing concentration of mag-fura-2. (**B**) Plot of extent of $Fe^{2+}$-mag-fura-2 complex formation (determined from data at 366 nm and expressed as fractional saturation, where one equals one $Fe^{2+}$ per initial [4Fe-4S] cluster) as a function of concentration of free mag-fura-2. The data were fitted using a simple binding isotherm, giving an apparent $K_d$, which reflects the competition between mag-fura-2 and RirA for iron. From this, the $K_d$ for binding of the fourth iron to the RirA cluster was determined as ~3 μM as described in Materials and methods.

DOI: https://doi.org/10.7554/eLife.47804.018

equilibrium, and that no further cluster (conversion/degradation) reaction occurred on the timescale of the measurement. Resulting spectra were corrected for contributions from the underlying FeS cluster absorbance and normalised, as described in Materials and methods, see *Figure 9*. The data describe the increasing proportion of $Fe^{2+}$ that is bound by the ligand and are fitted by a simple binding isotherm, yielding an apparent $K_d$ that reflects the competition between mag-fura-2 and [3Fe-4S] RirA for $Fe^{2+}$, from which the dissociation constant for RirA was readily determined as $K_d = 3 \pm 0.2$ μM. This value is entirely consistent with the $[4Fe-4S]^{2+} \leftrightarrow [3Fe-4S]^{0} + Fe^{2+}$ equilibrium functioning as the iron-sensing reaction.

## Iron-responsive cluster conversion in dimeric [4Fe-4S] RirA revealed by mass spectrometry

Conditions for the detection of the RirA dimer by ESI-MS were optimised (*Pellicer Martinez et al., 2017*) and time-resolved ESI-MS was used to follow cluster degradation in the RirA dimer under anaerobic conditions, see *Figure 10A*. The major species prior to addition of iron chelator were the [4Fe-4S]/[4Fe-4S] and [3Fe-4S]/[4Fe-4S] forms. In the dimer, the presence of two clusters in many cases precludes unambiguous identification of the intermediates because the distribution of iron and sulphur across the two clusters is unknown. Furthermore, there are many more possible inter-mediates of the dimer as each of the two clusters undergoes conversion/degradation. Nevertheless, the temporal behaviour of tentatively assigned species could be fitted to a basic model of cluster conversion in dimeric RirA, see *Figure 10—figure supplement 1*. Plots of relative abundance of [4Fe-4S]/[4Fe-4S], [3Fe-4S]/[3Fe-4S], and [2Fe-2S]/[2Fe-2S] forms and fits to the global model are shown in *Figure 10B*, with rate constants in *Supplementary file 2*. Additional plots of [3Fe-4S]/[4Fe-4S], [2Fe-2S]/[3Fe-4S], [apo]/[2Fe-2S] and [apo]/[apo] are shown in *Figure 10—figure supplement 2*. A plot comparing the time dependence of the EPR signal intensity due to $[3Fe-4S]^{1+}$ forms under aerobic and anaerobic conditions with the ESI-MS abundance of the [3Fe-4S]/[3Fe-4S] species is shown in *Figure 10—figure supplement 3*. As in the case of the monomer, the aerobic ESI-MS and EPR data are in good agreement, consistent with the susceptibility of the $[3Fe-4S]^{0}$ cluster to oxidation.

We note that [3Fe-3S]/[3Fe-4S] and [3Fe-3S]/[3Fe-3S] cluster intermediates were not clearly observed; signals at the predicted masses for these species were observed, but only at very low

intensities, indicating that these forms are reactive intermediates during the dimer cluster conversion process and therefore do not accumulate. Importantly, the rate constant derived from the global fit for the initial reversible loss of $Fe^{2+}$ is very similar to that obtained from analysis of the monomer state (0.31 compared to 0.30 min$^{-1}$, **Table 1** and **Supplementary file 2**). Finally, the extent of sulphur adduct formation was greater than observed in the monomer region under equivalent conditions. Thus, within the limits of what we are able to observe, the RirA dimer behaved similarly to the monomer, consistent with monomer species forming from dimers during the measurement.

## Conclusions

The global iron regulator RirA differs in many respects from the well characterised bacterial iron regulator Fur, and so a distinct model is needed to account for how it senses and responds to the intracellular iron status. Data from ESI-MS under non-denaturing conditions, including $^{34}S$ isotope exchange, along with EPR spectroscopy, have provided a highly detailed view of the [4Fe-4S] RirA cluster degradation reactions in response to low iron and $O_2$, another key environmental signal. In particular, the work further illustrates the remarkable potential of ESI-MS to provide time-resolved information on metallo-cofactor reactivity (that involves a change in mass), which is not available

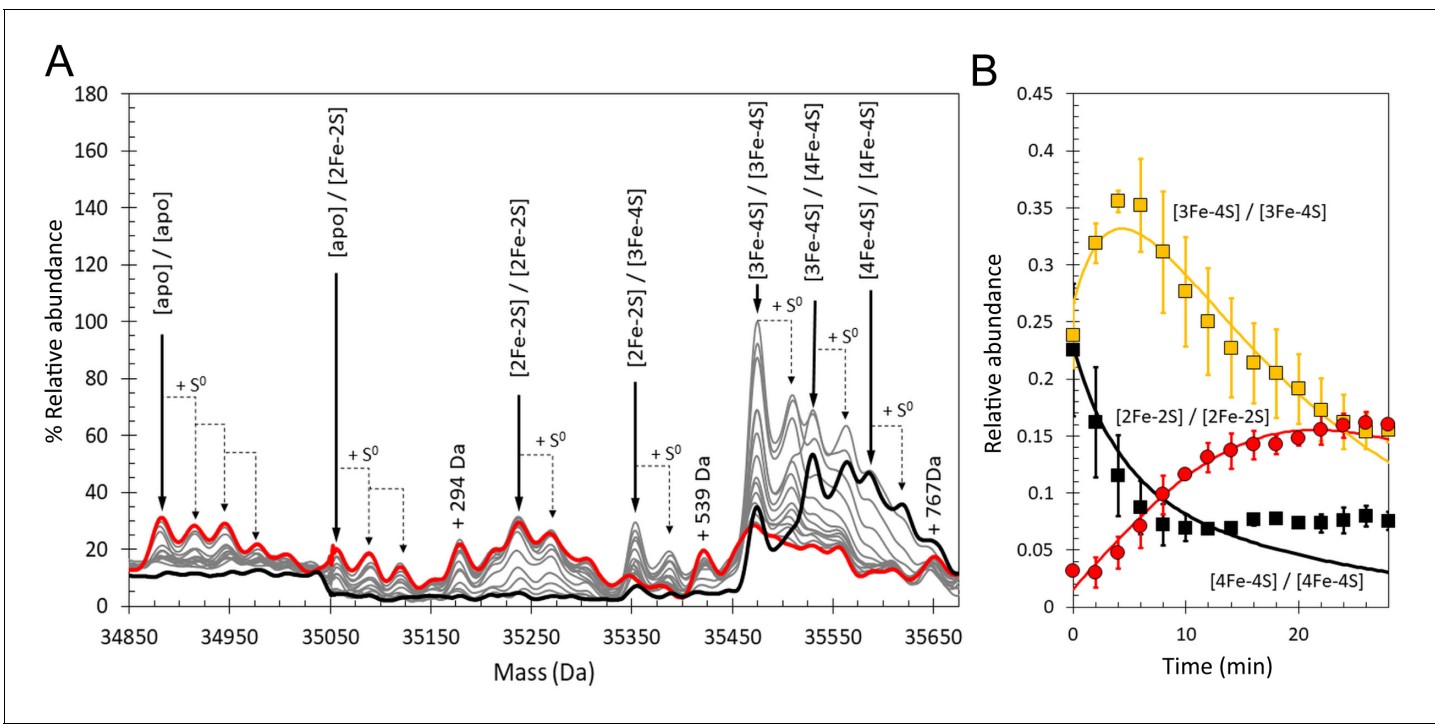

**Figure 10.** Time-resolved ESI-MS analysis of dimeric RirA cluster conversion under anaerobic, low iron conditions. (**A**) Deconvoluted mass spectrum of [4Fe-4S] RirA prior to the addition of 250 µM EDTA (black line) and 30 min after the addition at 37°C (red line) and at intervening time points (grey lines). [4Fe-4S] RirA (~25 µM) was in aerobic 250 mM ammonium acetate pH 7.3. Low iron conditions were generated by the addition of 250 µM EDTA and cluster conversion/degradation was followed at 37°C. The peak at +294 Da could represent an EDTA adduct of the apo-RirA dimer. (**B**) Plots of relative ESI-MS abundances of [4Fe-4S]/[4Fe-4S] (black), [3Fe-4S]/[3Fe-4S] (yellow), and [2Fe-2S]/[2Fe-2S] (red) forms of RirA as a function of time following exposure to 250 µM EDTA at 37° C. Fits of the data generated by a global analysis of the ESI-MS data based on the reaction scheme depicted in **Figure 10—figure supplement 1** are shown as solid lines. Error bars show standard error for ESI-MS dataset (n = 2 derived from two biological replicates).

DOI: https://doi.org/10.7554/eLife.47804.019

The following figure supplements are available for figure 10:

**Figure supplement 1.** Proposed reaction scheme for dimeric [4Fe-4S] RirA cluster conversion/degradation.
DOI: https://doi.org/10.7554/eLife.47804.020
**Figure supplement 2.** Kinetic analysis of dimeric [4Fe-4S] RirA cluster conversion/degradation.
DOI: https://doi.org/10.7554/eLife.47804.021
**Figure supplement 3.** Plot of ESI-MS relative abundances of [3Fe-4S]/[3Fe-4S].
DOI: https://doi.org/10.7554/eLife.47804.022

from other techniques. As shown in *Figure 5*, the mechanism involves a complex series of events initiated by the loss of iron, resulting in a [3Fe-4S] intermediate that decays further to [3Fe-3S], [3Fe-2S] and [2Fe-2S] species along the pathway to apo-RirA. Importantly, ESI-MS and absorbance data showed that the iron ($Fe^{2+}$) dissociation step occurs at a similar rate under both aerobic and anaerobic conditions, consistent with RirA functioning principally as an iron sensor.

Thus, the data presented here enabled us to elucidate at a molecular level the changes that occur in the transition from Fe-replete to Fe-depleted conditions. We propose a model in which the RirA [4Fe-4S]$^{2+}$ cluster continually undergoes $Fe^{2+}$ dissociation, to form a [3Fe-4S]$^{0}$ form, and, under iron replete conditions, re-association to re-form [4Fe-4S]$^{2+}$. When cellular iron becomes scarce, competition for the dissociated $Fe^{2+}$ increases, and re-association is disfavoured. In order to effectively sense cytoplasmic iron levels, the affinity (dissociation constant) of the fourth cluster iron must be in the range of normal free (chelatable) iron levels. Although unknown for *R. leguminosarum*, cytoplasmic free iron has been measured for *E. coli* and found to be ~10 µM (*Keyer and Imlay, 1996*). Assuming that free iron levels are not very different in *R. leguminosarum*, the $K_d$ of ~3 µM determined here for the [3Fe-4S]$^{0}$ + $Fe^{2+}$ ↔ [4Fe-4S]$^{2+}$ equilibrium is entirely consistent with an iron-sensing function. Indeed, we note that Fur, the iron-sensing regulator of *E. coli*, binds $Fe^{2+}$ with a $K_d$ of ~1–10 µM (*Bagg and Neilands, 1987*; *Mills and Marletta, 2005*).

Although done in vitro, the parameters that emerged are in keeping with its role as a Fe-responsive regulator in vivo. Here, as previously (*Pellicer Martinez et al., 2017*), we have used the iron chelator EDTA to simulate low iron, via its ability to coordinate $Fe^{2+}$ that has dissociated from the RirA [4Fe-4S] cluster. While EDTA is clearly not a physiological ligand, it nevertheless induces low iron conditions by providing a sink for iron that simulates conditions under which cluster degradation occurs. Conditions such as this must exist in the cytosol of cells growing under iron-limiting conditions, where a myriad of iron-requiring systems are in competition for available iron, and where a drop in 'free' iron levels results in loss of RirA-mediated transcriptional repression leading to up-regulation of iron uptake systems.

Competitive loss of $Fe^{2+}$ from [4Fe-4S]$^{2+}$ RirA results in a [3Fe-4S]$^{0}$ form that is unstable, undergoing degradation to an apo-form. This lability under low iron conditions is not a general property of [4Fe-4S] clusters, as previously demonstrated for the closely related [4Fe-4S] NsrR (*Pellicer Martinez et al., 2017*). This raises the question of why the RirA cluster is so fragile. One possibility is that the cluster is bound to the protein only by the three conserved Cys residues, with the tetrahedral coordination of the non-Cys bound iron completed by a weakly interacting ligand such as water or hydroxide. Such an arrangement, which would enhance the lability of the site-differentiated iron of the [4Fe-4S] cluster, would be similar to that found in the mammalian iron sensor protein IRE-BP (iron regulatory element-binding protein) (*Dupuy et al., 2006*), and it is tempting to speculate that this may be a general feature of iron-sensing iron–sulphur clusters. We also note that the key step of iron-sensing in RirA, the simple equilibrium [4Fe-4S]$^{2+}$ ↔ [3Fe-4S]$^{0}$ + $Fe^{2+}$, mirrors the iron-sensing step of the better known bacterial global iron regulators Fur and DtxR in which $Fe^{2+}$ binds the apo-form of these proteins (*Lee and Helmann, 2007*; *Pohl et al., 2003*; *D'Aquino et al., 2005*; *Ding et al., 1996*). However, the relative complexity of the RirA cofactor (co-repressor) enables it to also directly sense $O_2$.

A key step for $O_2$-sensing is the oxidation of [3Fe-4S]$^{0}$ to [3Fe-4S]$^{1+}$. The [3Fe-4S]$^{1+}$ intermediate was more reactive than the [3Fe-4S]$^{0}$ form, and several other subsequently formed intermediates also showed higher reactivity, leading to an overall enhanced rate of degradation (*Figure 5* and *Table 1*). The susceptibility of cluster iron and sulphide to oxidation in the presence of $O_2$ clearly drives the increased rate of cluster degradation. Thus, the [3Fe-4S]$^{0}$ cluster formed via the reversible dissociation of $Fe^{2+}$ from [4Fe-4S]$^{2+}$ RirA is susceptible to oxidation in the presence of $O_2$. We propose that, even under iron-replete conditions, in the presence of $O_2$ some oxidation of the RirA [3Fe-4S]$^{0}$ form occurs before it can re-associate with $Fe^{2+}$, resulting in cluster degradation.

Understanding how iron and $O_2$ signals are sensed and integrated is important for all organisms that depend on both iron and $O_2$. In the case of *R. leguminosarum*, RirA regulates not only iron uptake, but also iron–sulphur cluster biogenesis (*Todd et al., 2006*). The RirA $O_2$-sensing mechanism enables the cell to meet the requirement for iron–sulphur cluster biosynthesis under aerobic conditions (*Imlay, 2006*). Under low iron and in the presence of $O_2$, cluster degradation occurs more readily as a result of the combined effects of competition for iron and the enhanced rate of cluster degradation under oxidising conditions. The rate at which cluster conversion/degradation occurs is

relatively slow, particularly compared to the rate at which the [4Fe-4S] $\leftrightarrow$ [3Fe-4S] equilibrium is established. In some cases at least, sensing processes are relatively slow reactions. For example, the master regulator of the aerobic/anaerobic switch in many bacteria, [4Fe-4S] FNR, undergoes a similar [4Fe-4S] to [2Fe-2S] cluster conversion reaction over minutes (*Crack et al., 2008*; *Crack et al., 2007*), with in vivo transcriptional responses occurring over a similar timescale (*Partridge et al., 2007*). Responses to changes in iron levels mediated by Fur also occur over several minutes (*Pi and Helmann, 2017*). Thus, the cellular response to change does not necessarily need to occur instantly, perhaps reflecting that environmental changes themselves may occur over a period of time. The similarities between the FNR cluster conversion mechanism (*Crack et al., 2017*) and that described here for RirA are remarkable in that the two proteins have no sequence or structural similarity beyond the cluster, and FNR does not sense iron levels; its cluster is stable in the presence of iron chelators (*Crack et al., 2008*).

The advances reported here are relevant to all species that utilise RirA as a regulator, including several important pathogens of plants and animals. This work also contributes important new information about the widespread Rrf2 family of iron–sulphur cluster-binding regulators, and the variable ways in which they employ a cluster to sense environment change. Finally, it opens up new questions about RirA function, including: the nature of the cluster coordination; whether the $K_d$ for the [4Fe-4S]$^{2+}$ $\leftrightarrow$ [3Fe-4S]$^0$ + Fe$^{2+}$ equilibrium and the mechanism of cluster disassembly are affected by [4Fe-4S] RirA complex formation with DNA; which steps of the degradation pathway are key for loss of DNA-binding; and, whether disruption of one cluster is sufficient to modulate DNA-binding of the RirA dimer. Efforts to address these are currently underway.

## Materials and methods

### Key resources table

| Reagent type (species) or resource | Designation | Source or reference | Identifiers | Additional information |
|---|---|---|---|---|
| Gene (*Rhizobium leguminosarum* strain 8401) | *rirA* | NA | CAC35510 | |
| Peptide, recombinant protein | *R. leguminosarum* RirA | *Pellicer Martinez et al., 2017* | | |
| Peptide, recombinant protein | *Azotobacter vinelandii* NifS | *Crack et al., 2014b* | | The *nifS* plasmid was a kind gift from Prof. Dennis Dean (Virginia Tech) |
| Chemical compound, drug | $^{34}$S L-cysteine | *Crack et al., 2019* | | |
| Software, algorithm | Origin | Origin Lab | | Version 8 |
| Software, algorithm | DynaFit | BioKin Ltd; *Kuzmic, 1996* | RRID:SCR_008444 | Version 4 |

### Purification and reconstitution of [4Fe-4S] RirA

*R. leguminosarum* RirA was over-expressed in *E. coli* and purified as previously described (*Pellicer Martinez et al., 2017*). In vitro cluster reconstitution to generate [4Fe-4S] RirA was carried out in the presence of NifS, as described previously (*Crack et al., 2014b*). Protein concentrations were determined using the method of Bradford (Bio-Rad), with bovine serum albumin as the standard. Cluster concentrations were determined by iron and sulphide assays (*Crack et al., 2006*; *Beinert, 1983*) or by using an absorbance extinction coefficient at 383 nm for the RirA [4Fe-4S] cluster of 13,460 ± 250 M$^{-1}$ cm$^{-1}$ (*Pellicer Martinez et al., 2017* ). $^{34}$S labelled [4Fe-4S] RirA was generated by reconstitution using $^{34}$S-L-cysteine as previously described (*Crack et al., 2019*).

## Preparation of samples under low iron and low iron/aerobic conditions

[4Fe-4S] RirA was exchanged into anaerobic 250 mM ammonium acetate, pH 7.3, using a desalting column (PD-10, GE Healthcare) in an anaerobic glove box ($O_2$ < 2 ppm) and diluted to ~30 µM cluster using the same buffer. To simulate low iron conditions, the soluble high affinity iron chelator EDTA ($Fe^{2+}$-EDTA, log$K$ = 14.3, $Fe^{3+}$-EDTA, log$K$ = 25.1) (*Martell and Smith, 1974*) was added at 250 µM (final concentration) and the cluster response was followed via spectroscopy or mass spectrometry. To investigate the sensitivity of [4Fe-4S] RirA to low iron conditions in the presence of $O_2$, anaerobic protein samples in 250 mM ammonium acetate, pH 7.3, were rapidly diluted with an air-saturated buffer containing EDTA to give the desired final $O_2$ and EDTA concentrations and the sample was infused directly into the mass spectrometer via a syringe pump thermostatted at 37°C, or incubated at 37°C prior to rapid freezing in liquid nitrogen for EPR measurements. For absorbance experiments, 500 µM glutathione (GSH) was added to the sample to help solubilise oxidised sulphur species that otherwise caused significant scattering of light.

## Determination of Fe²⁺-binding affinity via competition assay

The dissociation constant ($K_d$) for $Fe^{2+}$ binding to [3Fe-4S]⁰ RirA was determined using an $Fe^{2+}$-binding competition assay employing the well-known divalent metal ligand mag-fura-2 (*Rodrigues et al., 2015*). Mag-fura-2 forms a 1:1 complex with $Fe^{2+}$, resulting in a shift of absorbance maximum from 366 nm (for metal-free mag-fura-2) to 325 nm (*Rodrigues et al., 2015*). Mag-fura-2 was dissolved in ultra-pure water to give a 2 mM stock solution and stored at −80°C until needed. Titrations of ~7 µM [4Fe-4S] RirA in 25 mM HEPES, 50 mM NaCl, 750 mM KCl, pH 7.5 with mag-fura-2 were carried out under anaerobic conditions at room temperature. Optimisation experiments showed that absorbance changes due to $Fe^{2+}$-binding to the Mag-fura-2 occurred quickly and so UV-visible spectra were recorded using a Jasco V500 spectrometer immediately following each addition of the chelator, such that the titration was complete within ~60 min. Overnight incubation of RirA with Mag-fura-2 led to cluster conversion/breakdown, and increased coordination of $Fe^{2+}$ to the chelator. Thus, the dissociation of a single $Fe^{2+}$ from the [4Fe-4S]²⁺ cluster occurred on a very different timescale to cluster conversion, permitting investigation of the [4Fe-4S] to [3Fe-4S] equilibrium.

A potential difficulty of using this or any ligand that gives rise to absorbance in the UV-visible region is that its absorbance spectrum overlaps that of the RirA cluster. Furthermore, the absorbance due to the cluster changes upon loss of a single $Fe^{2+}$, and so a robust method to correct for these changes was required. Fortunately, the absorbance profiles of apo- and divalent metal-bound mag-fura-2 overlap, with an isosbestic point at 346 nm, providing a means to correct for underlying absorbance changes due to the cluster (*Rodrigues et al., 2015*; *Simons, 1993*). Firstly, the spectrum due to initial [4Fe-4S] cluster was subtracted from each subsequent spectrum. Then the data were normalised so that the resulting nest of spectra reported on the changing proportions of $Fe^{2+}$-bound and apo-mag-fura-2. Finally, spectra were corrected for any changes in the subtracted cluster spectrum using a scaling factor that preserved the isosbestic point. Spectra of mag-fura-2 in apo and $Fe^{2+}$ forms (in the absence of RirA) were used to assist with this. Changes due to the scaling factor were <0.07 absorbance units, within the range expected for the absorbance change during cluster conversion. From this, the percentage of maximum $Fe^{2+}$-mag-fura-2 complex present (where 100% equated to the concentration of [4Fe-4S] RirA, that is one $Fe^{2+}$ per cluster) was plotted as a function of free mag-fura-2. This resulted in a plot with a hyperbolic form, which was fitted with an equation describing a simple binding isotherm using Origin8 (OriginLabs). This yielded an apparent competition dissociation constant from which the $K_d$ for RirA was determined using the expression:

$$K_{d(MF2\,app)} = K_{d(MF2)}\left(1 + \frac{[RirA]}{K_{d(RirA)}}\right)$$

Where $K_{d(MF2\,app)}$ is the determined apparent $K_d$ from the competition binding assay, $K_{d(MF2)}$ is the $K_d$ for mag-fura-2 binding to $Fe^{2+}$, (previously determined as $K_d$ = 2.05 µM; *Rodrigues et al., 2015*) and $K_{d(RirA)}$ is the $K_d$ for [3Fe-4S] binding to $Fe^{2+}$ (*Hulme and Trevethick, 2010*). The resulting $K_d$ and standard error was determined from three equivalent titration experiments.

## ESI-MS measurements

[4Fe-4S] RirA samples were infused directly (0.3 mL/h) into the ESI source of a Bruker micrOTOF-QIII mass spectrometer (Bruker Daltonics, Coventry, UK) operating in the positive ion mode, and calibrated using ESI-L Low Concentration Tuning Mix (Agilent Technologies, San Diego, CA). Mass spectra ($m/z$ 500–1750 for RirA monomer; $m/z$ 1,800–3,500 for RirA dimer) were acquired for 5 min using Bruker oTOF Control software, with parameters as follows: dry gas flow 4 L/min, nebuliser gas pressure 0.8 Bar, dry gas 180°C, capillary voltage 2750 V, offset 500 V, ion energy 5 eV, collision RF 180 Vpp, collision cell energy 10 eV. Optimisation of experimental conditions for the transmission of dimeric species was achieved by increasing the capillary voltage to 4000 V and the collision RF to 600 Vpp (*Laganowsky et al., 2013*).

Processing and analysis of MS experimental data were carried out using Compass DataAnalysis version 4.1 (Bruker Daltonik, Bremen, Germany). Neutral mass spectra were generated using the ESI Compass version 1.3 Maximum Entropy deconvolution algorithm over a mass range of 17,300–18,000 Da for the monomer and 34,850–35,810 Da for the dimer. For kinetic modelling, in order to clearly resolve overlapping peaks, multiple Gaussian functions were fitted to the experimental data using a least-squares regression function in Origin 8 (Origin Lab) (*Laganowsky et al., 2013*). Exact masses are reported from peak centroids representing the isotope average neutral mass. For apo-proteins, these are derived from m/z spectra, for which peaks correspond to $[M + nH]^{n+}/n$. For cluster-containing proteins, where the cluster contributes charge, peaks correspond to $[M + (Fe-S)^{x+} + (n-x)H]^{n+}/n$, where M is the molecular mass of the protein, Fe-S is the mass of the particular iron–sulphur cluster of x+ charge, H is the mass of the proton and n is the total charge. In the expression, the x+ charge of the iron–sulphur cluster offsets the number of protons required to achieve the observed charge state (n+) (*Johnson et al., 2000*). Predicted masses are given as the isotope average of the neutral protein or protein complex, in which iron–sulphur cluster-binding is expected to be charge-compensated (*Crack et al., 2017*; *Kay et al., 2016*).

Mass spectra are plotted as percentage relative abundances, where the most abundant species is arbitrarily set to 100% and all other species are reported relative to it. Time-resolved MS intensity data for global analysis was processed to generate relative abundance plots of ion counts for the relevant species as a fraction of the total ion count for all species. This permitted changes in relative abundance to be followed without distortions due to variations in ionisation efficiency that normally occur across a data collection run. Some variation in the starting spectrum was observed due to the presence of cluster breakdown products, which affected concentrations of intermediates during the cluster conversion/breakdown process; these variations are represented by error bars in the relative abundance plots of *Figures 4* and *7*. Such plots were analysed globally using the program Dynafit 4 (BioKin Ltd) (*Kuzmic, 1996*), which employs nonlinear least-squares regression of kinetic data, based on multi-step mechanisms, from which fits of the experimental data were generated and rate constants estimated, as previously reported (*Crack et al., 2017*). The presence of breakdown products in the starting spectrum was accounted for by allowing starting abundance to be offset from zero. Briefly, the kinetic model consisted of a series of sequential or branched reactions beginning with the dissociation of $Fe^{2+}$ from $[4Fe-4S]^{2+}$ to form $[3Fe-4S]^{0}$ and the reverse reaction, the binding of $Fe^{2+}$ to $[3Fe-4S]^{0}$ to reform the $[4Fe-4S]^{2+}$ cluster. With the exception of the latter, which is a second order process, all steps in the proposed mechanism are first order. All steps of the mechanism are shown in *Figure 5*.

## Spectroscopy

Absorbance kinetic data at $A_{386\ nm}$ were recorded via a fibre optic link, as previously described (*Crack et al., 2007*). EPR measurements were made with an X-band Bruker EMX EPR spectrometer equipped with a helium flow cryostat (Oxford Instruments). Unless stated otherwise, EPR spectra were measured at 10 K at the following instrumental settings: microwave frequency, 9.471 GHz; microwave power, 3.18 mW; modulation frequency, 100 kHz; modulation amplitude, 5 G; time constant, 82 ms; scan rate, 22.6 G/s; single scan per spectrum. Relative concentrations of the paramagnetic species were measured using the procedure of spectral subtraction with a variable coefficient (*Svistunenko et al., 2006*) and converted to absolute concentrations by comparing an EPR spectrum second integral to that of a 1 mM Cu(II) in 10 mM EDTA standard, at non-saturating values of the microwave power. The RirA EPR signal saturation was studied by taking EPR measurements at 13

values of microwave power, ranging from 0.2 µW to 200 mW, at eight temperature values, ranging from 4 K to 50 K. The EPR sample was equilibrated at every new temperature for at least 8 min before the power set of spectra measurements commenced.

## Acknowledgements

This work was supported by Biotechnology and Biological Sciences Research Council through grants BB/E003400/1, BB/J003247/1 and BB/L006140/1, by UEA through the award of a PhD studentship to MTPM and purchase of the ESI-MS instrument, and by the FeSBioNet COST Action CA15133.

## Additional information

### Funding

| Funder | Grant reference number | Author |
| --- | --- | --- |
| Biotechnology and Biological Sciences Research Council | BB/E003400/1 | Andrew WB Johnston<br>Nick E Le Brun |
| Biotechnology and Biological Sciences Research Council | BB/J003247/1 | Nick E Le Brun |
| Biotechnology and Biological Sciences Research Council | BB/L006140/1 | Jason C Crack<br>Nick E Le Brun |
| Horizon 2020 Framework Programme | CA15133 | Nick E Le Brun |

The funders had no role in study design, data collection and interpretation, or the decision to submit the work for publication.

### Author contributions

Ma Teresa Pellicer Martinez, Formal analysis, Investigation, Writing—original draft; Jason C Crack, Conceptualization, Data curation, Formal analysis, Supervision, Investigation, Methodology, Writing—original draft; Melissa YY Stewart, Data curation, Formal analysis, Investigation, Writing—original draft; Justin M Bradley, Supervision, Methodology, Writing—original draft; Dimitri A Svistunenko, Data curation, Formal analysis, Supervision, Investigation, Writing—original draft; Andrew WB Johnston, Conceptualization, Writing—original draft, Writing—review and editing; Myles R Cheesman, Conceptualization, Supervision, Writing—original draft; Jonathan D Todd, Conceptualization, Writing—review and editing; Nick E Le Brun, Conceptualization, Supervision, Funding acquisition, Investigation, Writing—original draft, Project administration, Writing—review and editing

### Author ORCIDs

Nick E Le Brun (ID) https://orcid.org/0000-0001-9780-4061

### Decision letter and Author response

Decision letter https://doi.org/10.7554/eLife.47804.031
Author response https://doi.org/10.7554/eLife.47804.032

## Additional files

### Supplementary files

• Supplementary file 1. Predicted and observed masses for apo- and cluster-bound forms of RirA.
DOI: https://doi.org/10.7554/eLife.47804.023

• Supplementary file 2. Rate constants resulting from global fit of experimental ESI-MS data for RirA dimer using the model shown in *Figure 10—figure supplement 1*.
DOI: https://doi.org/10.7554/eLife.47804.024

• Transparent reporting form

DOI: https://doi.org/10.7554/eLife.47804.025

## Data availability

All data generated or analysed during this study are included in the manuscript and supporting files. ESI-MS datasets have been deposited at Open Science Framework (http://doi.org/10.17605/OSF. IO/JMF6H). Source data files are available on the Open Science Framework (http://doi.org/10. 17605/OSF.IO/H2M4P).

The following datasets were generated:

| Author(s) | Year | Dataset title | Dataset URL | Database and Identifier |
|---|---|---|---|---|
| Ma Teresa Pellicer Martinez, Jason C Crack, Melissa YY Stewart, Nick E Le Brun | 2019 | ESI-MSanalysis of RirA under anaerobic low iron conditions: monomer region | http://doi.org/10.17605/ OSF.IO/JMF6H | OSF, 10.17605/OSF. IO/JMF6H |
| Ma Teresa Pellicer Martinez, Jason C Crack, Melissa YY Stewart, Justin M Bradley, Dimitri A Svistunenko, Andrew WB Johnston, Myles R Cheesman, Jonathan D Todd, Nick E Le Brun | 2019 | RirA ESI-MS and EPR study | http://doi.org/10.17605/ OSF.IO/H2M4P | OSF, 10.17605/OSF. IO/H2M4P |

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
