## [Decision Letter]

[Editors’ note: a previous version of this study was rejected after peer review, but the authors submitted for reconsideration. The first decision letter after peer review is shown below.]

Thank you for submitting your work entitled "Mechanisms of iron- and O_2_-sensing by the [4Fe-4S] cluster of the global iron regulator RirA" for consideration by *eLife*. Your article has been reviewed by a Senior Editor, a Reviewing Editor, and three reviewers. The reviewers have opted to remain anonymous.

Our decision has been reached after extensive consultation between the reviewers. The reviewers' concerns are summarized below. We regret to inform you that in its present form, your work can not be considered for publication in *eLife*. However, since all reviewers agreed that the work is technically elegant, innovative and exciting, we would be willing to consider a resubmission that addresses all concerns raised.

Summary:

The manuscript by Pellicer Martinez et al., employs a combination of non-denaturing ESI-MS with more established techniques including UV-Vis absorption and EPR spectroscopy to elucidate the breakdown of the iron- and oxygen-sensitive [4Fe-4S] cluster from the global iron sensor RirA from Rhizobium.

The authors propose mechanisms of how RirA might sense iron (directly) and oxygen (indirectly) via different cluster degradation states.

The potential impact of this work stems from two angles: (1) providing insight into iron sensing in an agronomically important bacterium (and beyond, perhaps including mammalian IRE-BP), and (2) the sophisticated methodology employing non-denaturing ESI-MS with alternative stable isotopes that allow unambiguous assignment of cluster composition, at least in the monomer. Whilst neither using stable isotopes nor conducting kinetic experiments by ESI-MS are new, their application to iron-sulfur clusters, and using 34S, is innovative (as also demonstrated in a few recent papers by the same authors), and is likely to stimulate similar studies for other Fe-S proteins, where much remains to be elucidated.

The reviewers acknowledge that new FeS cluster states have been identified convincingly. However, a number of problems were identified, and are listed and elaborated below.

Essential revisions:

1) A convincing link of at least some of the (new) cluster states to the in vivo situation is missing.

The authors propose that Fe(II) reversibly dissociates from the [4Fe-4S] cluster leading to [3Fe-4S]^0^, and that this equilibrium is a key sensing mechanism for iron in the cell.

a) A key missing piece of information to support this mechanism is the affinity constant for the fourth iron (or indeed the seventh and eighth iron in the dimer). This could be determined, e.g., using a chelator competition assay (refer for an example to Rodrigues et al., 2015; they use Mag-fura-2, a metal chelator to determine iron equilibrium by UV-vis spectroscopy). The binding constant would then be expected to be in the range of cellular free iron.

b) A comment as to the suitability of EDTA to provide conditions of "low iron" would also be desirable. Is it thought that EDTA may abstract Fe(II) directly form the cluster, or are there two complexation equilibria at play? Are the conditions comparable to the bacterial cytosol?

c) Related to these issues: Is a [3Fe-4S] cluster RirA really binding with different affinity to DNA? In ref. (27), only [4Fe-4S], [2Fe-2S] and apo-forms were compared, and the [2Fe-2S] form bound (only) ca. 5-fold weaker.

Therefore, an affinity/dissociation constant for Fe(II) needs to be determined, and the proposed sensing mechanism needs to be put into physiological context.

2) What is the significance of the reported kinetic data?

The data are not only quite complex and require more elaborate explanations, but also need to be put into overall context.

a) There is a need to clarify the kinetic model(s) underlying the fits (first order/pseudo-first order?). The stoichiometry and mechanistic assumptions need to be laid out for each set of MS data fitted. This includes appropriate appreciation of involvement of monomers and dimers.

b) The magnitude of the reaction rates/kinetic constants: How do these relatively slow rates for cluster conversion relate to the physiological context? Why are the data reported in Pellicer Martinez et al., 2018 and those reported in the present manuscript so different (2 orders of magnitude)?

c) Furthermore, it appears that EDTA is added to mimic "iron-depleted conditions" – but can a direct removal of Fe by EDTA be excluded? This would require demonstrating independence of the reaction rates from EDTA concentration, or observing dissociation as a consequence of diluting holo RirA into buffer (which might perhaps also allow the determination of a dissociation constant).

d) Given that certain processes turn out to be affected by oligomerisation state (see subsection “Iron-responsive cluster conversion in dimeric [4Fe-4S] RirA revealed by mass spectrometry”), and that dimers will predominate in solution, it appears that the absorbance and EPR kinetic data should also be compared to the ESI-MS kinetic data of the dimer, inasmuch as this is possible (e.g. loss of the first two Fe(II) ions).

e) A related problem concerns the conflation of kinetics and thermodynamics. How may one conclude that there is no cooperativity when k(forward) is the same, but k(reverse) is larger in the dimer? Since KD = k(forward)/k(reverse) (for dissociation), it seems that binding of the "eighth" Fe(II) in the dimer would be stronger than in the monomer? Or could this be a consequence of the ESI conditions? Also – what does it mean when k2 and k-2 are equal? Perhaps it would be best not to invoke "cooperativity" at all (at least not with respect to ESI-MS results), as the monomers observed by ESI-MS are (mainly) generated from (reacted) dimers by the ESI process, rather than being already present in solution. In contrast, the Fe(II) dissociation is a process that occurs in solution, where the dimers dominate. These considerations also highlight that the significance of monomers and dimers has not yet been addressed thoroughly.

In summary, this part of the interpretation and discussion needs more thought, and may also be helped tremendously by the determination of a Fe(II) binding/dissociation constant requested above.

In relation to this part of the discussion, the authors also need to be careful with terms like "stable", "unstable", "stability" – these are thermodynamic, not kinetic terms. Perhaps "reactive" or "labile" would be more appropriate; but above all, the authors must take care to distinguish clearly between kinetics and thermodynamics.

3) The proposed mechanism of oxygen sensing via oxidation of the [3Fe-4S] cluster and accelerated degradation is not clearly and convincingly elucidated

It is proposed (mainly by EPR) that oxygen is sensed via oxidation to the [3Fe-4S]^1+^ cluster intermediate. An EPR active species was observed at time zero (g = 2.01) and was assigned as the [3Fe-4S]^1+^ intermediate based on temperature and power dependencies (Figure 8). In the presence of oxygen the signal increased and decayed over time. This species, however, is present at the beginning of the experiment AND at the end in both aerobic and anaerobic conditions (Figure 8A,D). This unstable species should completely disappear under aerobic conditions (as predicted by the theoretical orange fitting curve in Figure 8C; overall poor fit!). Therefore, we are concerned that this initial signal at g = 2.01 is a (stable) species that may not necessarily be part of the reaction mechanism. Without error bars for the EPR data under anaerobic conditions in Figure 8C, it is hard to tell if the supposed intermediate actually increases with time.

The authors suggest that the initial difference in abundance of the [3Fe-4S] species between MS and EPR in Figure 8C is due to the [3Fe-4S]^0^ species, which is EPR inactive (lines Subsection “The first step of 4Fe-4S] RirA cluster conversion is the O_2_-independent reversible loss of Fe^2+^ to form [3Fe-4S]”). Can it be ruled out that the lack of EPR signal is simply due to the regulator converting from the [4Fe-4S]^2+^ state to the [3Fe-4S] state, in an oxygen dependent manner? This reaction would also explain the observation of the resolved buildup in the first couple of minutes of the [3Fe-4S]^1+^ species by EPR. Does Fe dissociate from the [4Fe-4S] also in the absence of chelator? An explanation for the difference with MS could be that due to the ionization conditions of ESI-MS, temporal resolution of the initiation of the reaction is not possible. Without this resolution, the kinetic analysis that results from the speciation diagrams in Figure 4 and Figure 7 could be difficult to interpret. As displayed in Figure 2A, the ionization conditions of ESI-MS lead to many Fe-bound species at t = 0. Can the authors exclude that small amounts of oxygen at the injection port of the MS or in the dry gas may lead to the observed cluster decomposition at t = 0? Comparing Figure 2A and Figure 6A, there is sample variation before the addition of chelator, which may be the result of small amounts of oxygen. O_2_ levels can be much better controlled in EPR sample preparation. Therefore overall, we feel it cannot be definitely concluded that O_2_ is reacting with the [3Fe-4S]^0^ species and not the [4Fe-4S] cluster.

[Editors’ note: what now follows is the decision letter after the authors submitted for further consideration.]

Thank you for submitting your article "Mechanisms of iron- and O_2_ -sensing by the [4Fe-4S] cluster of the global iron regulator RirA" for consideration by *eLife*. Your article has been reviewed by Gisela Storz as the Senior Editor, a Reviewing Editor, and three reviewers. The following individuals involved in review of your submission have agreed to reveal their identity: David P. Giedroc (Reviewer #3).

The reviewers have discussed the reviews with one another and the Reviewing Editor has drafted this decision to help you prepare a revised submission.

Summary:

In this manuscript, Pellicer Martinez and coworkers provide elegant mechanistic studies on how RirA of Rhizobium may function as a global iron regulator through the decay of its [4Fe-4S] cluster. By using an assortment of spectroscopic and spectrometric methods, global reaction time courses for the conversion of various Fe/S cluster species are provided.

The work is significant in that it provides an impressive case study for how gentle ionization (ESI) or "native-state" mass spectrometry can be leveraged to learn something about intermediates along a reaction coordinate in a metal-sensing regulatory protein. Two key additional experiments substantially increase information content: (1) parallel experiments monitored by EPR under the same solution conditions; and (2) the dissociation of the first Fe from the cluster occurs with an equilibrium constant that is consistent with this equilibrium being "tuned" to the bioavailable Fe in the bacterial cell. The first is important in that minimizes a commonly raised criticism of (all) ESI-MS experiments like these, namely that the ionization process itself and injection into the gas phase, is perturbing the system. The second provides new insights into how one connects cluster disassembly to Fe-sensing in cells.

The authors further propose that RirA can additionally sense oxygen levels as evidenced by the increased rate of decay of an intermediate [3Fe-4S] species in the presence of oxygen.

The experimental data are sound and well controlled in the manuscript. This work is not the first where native ESI-MS has been applied to Fe-S clusters, and it is also not the first study on RirA either. Overall, the authors provide a nice follow-up study to their 2017 Chemical Science paper. RirA is thought to be operating in organisms that lack the conventional Fe-sensor, Fur, and thus detailed study of this metal sensor allows comparisons to previous (albeit less comprehensive) studies of Fur. This work also provides new regulatory insights into a member of the enigmatic Rrf2 family of putative oxygen and Fe-sensing regulators, which are generally lacking. We conclude that the topic as such and this work in particular is worthy of inclusion in *eLife*, based on quality, rigour, interest, and novelty. Future studies, especially work in vivo, will have to confirm the predictions from this in vitro study.

As far as the authors' responses to the joint reviewers' letter are concerned, we conclude that the concerns raised in the letter have been largely addressed. Remaining issues as listed below should be addressed in a (minor) revision.

Essential revisions:

Abstract: The phrase "…for which K_d_ = ~3 μM, consistent with this equilibrium sensing 'free' iron in the cell cytoplasm." sounds awkward and is not connected to the first part of the sentence. Further, it is not really explained in the Abstract how the authors envision iron and oxygen regulation. Please re-write and make these points clear.

Figure 4 and Figure 7: It is not fully clear to us why an apo species appears in Figure 4E (at least 70%) and 7F (plateau at 90%) but the A(382nm) does not decay substantially. As far as we understand, the cluster in MS analysis is gone after 30 min (all MS-detectable species are close to zero), but the colour at 382 nm remains. Please clarify and explain in text. Minor point: It may be didactically better to present Figure 4 and Figure 7 in the very same order.

It is also unclear how the presence of the initially present cluster breakdown products affects the kinetic data and their modelling. The authors state that "The variation in the starting spectrum in terms of relative intensities does not affect the relative intensity behaviour of species as a function of time, as these are entirely reproducible." This is hard to comprehend – how can what is there at the start of a reaction not affect subsequent reactions? Perhaps the answer lies in what "the relative intensity behaviour" reflects and how it is used to arrive at the quantitative data, but this complication needs to be made clearer in the manuscript.

Figure 5: The first equilibrium is shown for [4Fe-4S]^2+^ going to a [3Fe-4S]^1+^. Wouldn't it be more appropriate to show the [3Fe-4S]^0^ species as the product of this first equilibrium, which then is converted to [3Fe-4S]^1+^ in the presence of oxygen?

Figure 1—figure supplement 1: The figure shows four highlighted Cys residues but only three are expected to be cluster coordinating. The homology model predicts that Cys17 is not close enough for cluster binding. This should be stated/clarified. It could also be stated that in the NsrR structure, Asp8 is the fourth coordinating residue. This Asp is not conserved in RirA.

One of the major previous concerns was the relation of the reported experiments to physiological context. In response, as requested by the reviewers, a dissociation constant for Fe(II) has been determined, even though – as the authors explain – they were concerned that this might not be possible due to the fragility of the [3Fe-4S]^0^ cluster formed in the initial "sensing" reaction. Surprisingly, the equilibrium titration worked nonetheless, with only ca. 1 Fe transferred to Mag-fura-2. An explanation why the subsequent cluster decay did, therefore, not happen (as the liberated Fe would also be picked up by the chelator) within the time frame of the experiment is missing. The time frame is mentioned but not specified. Was the titration done on a single solution, where subsequent additions of Mag-fura-2 were added to the same solution? Or does each point correspond to a different solution?

Further major concerns were related to erroneous kinetic constants – the ensuing corrections have addressed these concerns to a significant degree. This includes questions around the effects of oxygen.

However, in the discussion regarding the identity of the various adducts, the authors mention that O_2_ can be generated in the ESI-MS source, and that this could be the origin of the adducts with oxygen. It is important to acknowledge this, and to clarify whether or not this phenomenon affects the results.

The question how the (relatively slow) kinetics observed in the current work relate to the processes occurring in vivo has been addressed in the response letter, but not necessarily in the manuscript. The conclusions mention continuous dissociation and association, but considering that kinetics are at the heart of the paper, it would be nice to have a comment on timescales here.

Finally, although the authors do a beautiful job exploring the disassembly of the 4Fe-4S cluster in RirA free in solution, they do not consider the impact that the bound DNA might have, in any way. If this is a true sensing mechanism then DNA binding by the repressing 4Fe-4S complex may well impact both the rate of dissociation of the "fourth Fe" as well as its equilibrium affinity (or not). Although we realize that such an investigation is well beyond the scope of this manuscript, simply ignoring this essential aspect of this system is less than satisfactory. We suggest the authors work a bit of this into their conclusions. Is it known whether the metallated RirA-DNA complex can be studied by native ESI-MS?

---

## [Author Response]

[Editors’ note: the author responses to the first round of peer review follow.]

Essential revisions:1) A convincing link of at least some of the (new) cluster states to the in vivo situation is missing.The authors propose that Fe(II) reversibly dissociates from the [4Fe-4S] cluster leading to [3Fe-4S]^0^, and that this equilibrium is a key sensing mechanism for iron in the cell.a) A key missing piece of information to support this mechanism is the affinity constant for the fourth iron (or indeed the seventh and eighth iron in the dimer). This could be determined, e.g., using a chelator competition assay (refer for an example to Rodrigues et al., 2015; they use Mag-fura-2, a metal chelator to determine iron equilibrium by UV-vis spectroscopy). The binding constant would then be expected to be in the range of cellular free iron.

We thank the reviewers for the comment and for the suggestion of mag-fura-2 as a potentially useful ligand. We had not previously attempted to try to measure the affinity of the fourth iron for the cluster because we originally thought that the fragility of the cluster/subsequent breakdown of the [3Fe-4S] form would prevent us from doing this. However, following the reviewers’ advice, we have succeeded in determining the *K*_d_ for the [4Fe-4S]^2+^ ↔ [3Fe-4S]^0^ + Fe^2+^ equilibrium.

A potential problem of using mag-fura-2 as a competitive ligand is that its absorbance spectrum overlaps that of the RirA cluster (250 – 450 nm) (Pellicer Martinez et al., 2017). Given that the absorbance due to the cluster changes during the cluster conversion process, beginning with the [4Fe-4S] to [3Fe-4S] equilibrium step, a method to correct for these changes was needed. Fortunately, both apo- and Fe^2+^-Mag-Fura-2 absorb and a titration results in an isosbestic point at 346 nm, as illustrated in Rodriguez et al.,2015, providing a convenient and robust means to correct for small underlying absorbance changes.

A series of titrations of [4Fe-4S] RirA with mag-fura-2 was carried out. To analyse the data, the spectrum due to initial [4Fe-4S] cluster was first subtracted from each subsequent spectrum. Then the data were normalised so that each spectrum simply reports on the proportions of Fe^2+^-bound and apo-mag-fura-2 present, and then corrected for any changes in the subtracted cluster spectrum by applying a correction factor to maintain the isosbestic point (Figure 9 in the revised manuscript). From this, the concentration of Fe^2+^-mag-fura-2 was determined at each point in the titration. Importantly, the concentration of Fe^2+^-mag-fura-2 plateaued at an equivalent of ~1 iron per cluster. This showed that one iron per cluster was readily available for complexation, confirming that the chelator experiments reported only on the [4Fe-4S] to [3Fe-4S] equilibrium reaction (a requirement for determination of the *K*_d_). A plot of percentage of maximum Fe^2+^-mag-fura-2 complex observed as a function of free mag-fura-2 concentration was generated. This had the form of a hyperbolic function and could be fitted as a simple binding isotherm using Origin8 (OriginLabs), yielding an apparent dissociation constant. This, of course, represents the competition between mag-fura-2 and [3Fe-4S] RirA for Fe^2+^, and the *K*_d_ for RirA was determined from the expression:

KdMF2app=KdMF2(1+RirAKdRirA)

This gave a *K*_d_ of 3 ± 0.2 µM (n = 3). Although the concentration of ‘free’ iron has not been reported for *R. leguminosarum*, it has been shown to be ~10 µM in *E. coli* (Keyer and Imlay, 1996), and Fur, the iron-sensing regulator of *E. coli*, binds Fe^2+^ with a *K*_d_ of ~1 – 10 µM (Bagg and Neilands, 1987; Mills and Marletta, 2005). There is no a priori reason to think that the values should be very different in other proteobacteria, including *Rhizobium,* so the *K*_d_ measured here for RirA is entirely consistent with the [4Fe-4S]^2+^ to [3Fe-4S]^0^ equilibrium being the RirA iron-sensing step.

b) A comment as to the suitability of EDTA to provide conditions of "low iron" would also be desirable. Is it thought that EDTA may abstract Fe(II) directly form the cluster, or are there two complexation equilibria at play? Are the conditions comparable to the bacterial cytosol?

We have shown previously (Pellicer Martinez et al., 2017) that the rate of cluster degradation under anaerobic conditions showed no significant dependence on the concentration of EDTA. Furthermore, exposure to Chelex-100 resin separated from the protein by a semi-permeable membrane resulted in cluster conversion at a similar rate. Thus, low ‘free’ iron conditions are generated via the ability of iron chelators to coordinate Fe^2+^ that has dissociated from the RirA [4Fe-4S] cluster (i.e. it involves two complexation equilibria: dissociation of Fe^2+^ from the cluster and then chelation of that Fe^2+^ by the competing ligand (EDTA in this case). While EDTA is clearly not a physiological ligand, it nevertheless induces low iron conditions by providing a sink for iron, and thus simulates conditions under which cluster degradation occurs. Conditions such as this must exist in the cytosol under iron-limiting conditions, where a myriad of iron-requiring systems are in competition for available iron. A drop in ‘free’ iron levels results in degradation of the RirA cluster and up-regulation of iron uptake systems. We now describe the context of using EDTA in more detail in the revised manuscript both at first mention of EDTA and in the Conclusions section.

c) Related to these issues: Is a [3Fe-4S] cluster RirA really binding with different affinity to DNA? In ref. (27), only [4Fe-4S], [2Fe-2S] and apo-forms were compared, and the [2Fe-2S] form bound (only) ca. 5-fold weaker.Therefore, an affinity/dissociation constant for Fe(II) needs to be determined, and the proposed sensing mechanism needs to be put into physiological context.

In fact, we don’t formally propose that [3Fe-4S] RirA has a significantly different DNA-binding affinity than [4Fe-4S] RirA; it might, but we cannot stabilise it sufficiently to test it. The key point is that once [3Fe-4S] is formed under low iron conditions (or in presence of O_2_), it undergoes further conversion/degradation to the [2Fe-2S] and subsequently apo- forms, which we have shown bind DNA less tightly ([2Fe-2S]) or not at all (apo). So, even though the [4Fe-4S] to [3Fe-4S] is the iron sensing step, the subsequent degradation of the cluster is required for the transduction of the signal. As explained above, we have determined the *K*_d_ for the binding of the fourth cluster iron and found that it is in the expected range for sensing free iron levels. We think that this and the revised discussion helps to frame the results reported here in a physiological context.

2) What is the significance of the reported kinetic data?The data are not only quite complex and require more elaborate explanations, but also need to be put into overall context.a) There is a need to clarify the kinetic model(s) underlying the fits (first order/pseudo-first order?). The stoichiometry and mechanistic assumptions need to be laid out for each set of MS data fitted. This includes appropriate appreciation of involvement of monomers and dimers.

The kinetic model (illustrated in Figure 5) was based on a series of sequential/branched reactions beginning with the dissociation of Fe^2+^ from [4Fe-4S]^2+^ to form [3Fe-4S]^0^ and the reverse reaction, the binding of Fe^2+^ to [3Fe-4S]^0^ to reform the [4Fe-4S]^2+^ cluster. This is the only second order step in the proposed mechanism; all other steps are first order. The kinetic data were analysed in terms of the kinetic model using the program DynaFit, which employs nonlinear least-squares regression analysis. Further explanation of the modelling is now given in the revised *Methods* section. We note that there was an error in Table 1, where the rate constant for the re-binding of Fe^2+^ to [3Fe-4S]^0^ was mis-reported. Our apologies for this error, which has now been corrected.

The significance of the kinetic data is that they reveal the mechanism of cluster conversion in detail. We would argue that the absolute rate constants determined are not so important, because the rate of reaction appears to be quite sensitive to the reaction conditions, e.g., buffer/pH, which most likely reflects the stability of the RirA protein and the lability of its cluster. However, the relative magnitudes of rate constants, which permit the observation of intermediate species, are important. Here, under a particular set of buffer conditions that permitted kinetic analysis by ESI-MS, UV-visible absorbance and EPR under identical conditions, the data show that the first step of the cluster degradation reaction is not affected by O_2_.

The ESI-MS data fitted here relate only to the monomeric form of RirA. Even though this was generated from the dimer during the measurement, no additional information relating to the dimer is contained with the data and so only the monomer form appears in the model. As described in the manuscript, we were also able to detect the dimeric form of RirA by ESI-MS (because the dissociation into monomers is partial). Although the kinetic modelling was limited by what can be assigned unambiguously, it is significant and consistent that the rate constants derived from the dimer data are similar to those obtained from the analysis of monomer (Table 1 and Supplementary file 2). Thus, there is no evidence for cooperativity between monomers of the dimer. However, because of the limited analyses of the dimer data, we take the reviewers’ later advice and do not address the issue of cooperativity in this manuscript.

b) The magnitude of the reaction rates/kinetic constants: How do these relatively slow rates for cluster conversion relate to the physiological context? Why are the data reported in reference 27 and those reported in the present manuscript so different (2 orders of magnitude)?

Reaction rates are indeed relatively slow, but in sensing processes, this is not particularly unusual. For example, the O_2_ sensor FNR also undergoes a similar [4Fe-4S] to [2Fe-2S] cluster conversion reaction over minutes (shown in vitro and in vivo) (Crack et al., 2007; Crack et al.,2008) and so it is clear that response to change does not occur instantly. Perhaps this is because environmental changes themselves may occur over a period of time. The difference between rates measured here and those previously reported are in part addressed in our response to comment 2(a); under different buffer conditions, different rates are observed. Importantly, though, here we have measured rates by ESI-MS and absorbance (as a control) to ensure that these techniques are reporting on the same reaction. Why is the reaction rate higher in ammonium acetate? It is not clear, but the RirA cluster is clearly more labile in the ammonium acetate buffer. One factor that might be important is that acetate is weakly coordinating.

c) Furthermore, it appears that EDTA is added to mimic "iron-depleted conditions" – but can a direct removal of Fe by EDTA be excluded? This would require demonstrating independence of the reaction rates from EDTA concentration, or observing dissociation as a consequence of diluting holo RirA into buffer (which might perhaps also allow the determination of a dissociation constant).

This is addressed in the answer to 1(b) above. We previously reported (Pellicer Martinez et al., 2017) that the rate of cluster degradation under anaerobic conditions showed no significant dependence on the concentration of EDTA, and a similar rate constant for cluster degradation to that observed with EDTA was measured in the presence of Chelex-100 resin (separated from the protein by a semi-permeable membrane), ruling out a direct interaction between the RirA cluster and EDTA. Indeed, the first indication of the lability of iron from the RirA [4Fe-4S] cluster came from gel filtration experiments (Pellicer Martinez et al., 2017) in which the amount of [3Fe-4S]^1+^ present after the column was significantly greater than before (from undetectable to ~13% total cluster), indicating that dilution of RirA and separation from small molecules results in increased loss of a single iron from the cluster. Thus, data relevant to these questions are already available. Furthermore, the *K*_d_ for this equilibrium has now been determined, see answer to 1(a).

d) Given that certain processes turn out to be affected by oligomerisation state (see subsection “Iron-responsive cluster conversion in dimeric [4Fe-4S] RirA revealed by mass spectrometry”), and that dimers will predominate in solution, it appears that the absorbance and EPR kinetic data should also be compared to the ESI-MS kinetic data of the dimer, inasmuch as this is possible (e.g. loss of the first two Fe(II) ions).

Modelling of the RirA dimer ESI-MS data is difficult because of the described problem of assigning cluster species when more than one cluster is present. We have done this as best we can and the main conclusion arising from it is that there is not much difference between the monomer and dimer kinetic data. This is consistent with the monomer arising from dissociation of the dimer during ionisation. The only straightforward direct comparison with the dimer data that can be made is between [3Fe-4S] EPR data and the [3Fe-4S]/[3Fe-4S] species. This is now included as Figure 10—figure supplement 3. The absorbance data are also consistent in that they indicate that conversion/degradation occurs over the same timescale as observed for dimeric RirA species.

e) A related problem concerns the conflation of kinetics and thermodynamics. How may one conclude that there is no cooperativity when k(forward) is the same, but k(reverse) is larger in the dimer? Since KD = k(forward)/k(reverse) (for dissociation), it seems that binding of the "eighth" Fe(II) in the dimer would be stronger than in the monomer? Or could this be a consequence of the ESI conditions? Also – what does it mean when k2 and k-2 are equal? Perhaps it would be best not to invoke "cooperativity" at all (at least not with respect to ESI-MS results), as the monomers observed by ESI-MS are (mainly) generated from (reacted) dimers by the ESI process, rather than being already present in solution. In contrast, the Fe(II) dissociation is a process that occurs in solution, where the dimers dominate. These considerations also highlight that the significance of monomers and dimers has not yet been addressed thoroughly.In summary, this part of the interpretation and discussion needs more thought, and may also be helped tremendously by the determination of a Fe(II) binding/dissociation constant requested above.

Because the monomer ESI-MS data can’t provide any information on cooperativity, and the dimer kinetic modelling is limited by the difficulty with unambiguous assignment, we agree that it is probably best not to discuss cooperativity in the manuscript. So, we have removed the sentence relating to this in the original version. Additionally, we apologise that there were errors in Table 2 (these resulted from a combination of a transposition error and using values from an earlier version of the modelling). These are now corrected, including *k*_-1_ and *k*_-2_ as second order rate constants.

In relation to this part of the discussion, the authors also need to be careful with terms like "stable", "unstable", "stability" – these are thermodynamic, not kinetic terms. Perhaps "reactive" or "labile" would be more appropriate; but above all, the authors must take care to distinguish clearly between kinetics and thermodynamics.

This is a perfectly correct point, for which we thank the reviewers. We have addressed this in the way suggested in the Discussion/conclusions section of the revised manuscript.

3) The proposed mechanism of oxygen sensing via oxidation of the [3Fe-4S] cluster and accelerated degradation is not clearly and convincingly elucidatedIt is proposed (mainly by EPR) that oxygen is sensed via oxidation to the [3Fe-4S]^1+^ cluster intermediate. An EPR active species was observed at time zero (g = 2.01) and was assigned as the [3Fe-4S]^1+^ intermediate based on temperature and power dependencies (Figure 8). In the presence of oxygen the signal increased and decayed over time. This species, however, is present at the beginning of the experiment AND at the end in both aerobic and anaerobic conditions (Figure 8A,D). This unstable species should completely disappear under aerobic conditions (as predicted by the theoretical orange fitting curve in Figure 8C; overall poor fit!). Therefore, we are concerned that this initial signal at g = 2.01 is a (stable) species that may not necessarily be part of the reaction mechanism. Without error bars for the EPR data under anaerobic conditions in Figure 8C, it is hard to tell if the supposed intermediate actually increases with time.

It is already established that the RirA [4Fe-4S] cluster is prone to loss of a single iron during sample manipulation and so the detection of a small component (~4%) of cluster in a [3Fe-4S]^1+^ form (and some also present in the EPR-silent [3Fe-4S]^0^ form) was not unexpected. That this signal does not significantly increase under anaerobic conditions following exposure to an iron chelator is consistent with the absence of an oxidant that would generate the EPR active form. That the signal does not entirely decay away, remaining at ~4% after 21 mins, was also not expected initially, but we have seen this kind of behaviour before, with the O_2_-sensing regulator FNR, where the EPR-active intermediate also does not entirely decay away to zero (unpublished observations). It is likely that a small proportion of cluster gets stuck in the [3Fe-4S]^1+^ form, as an off-pathway product. However, this does not affect at all the main conclusion of these experiments – that the presence of O_2_ results in significant oxidation of the [3Fe-4S]^0^ form to [3Fe-4S]^1+^. This is an indisputable conclusion from the EPR data. Nevertheless, we have now addressed the lingering [3Fe-4S]^1+^ signal in the manuscript.

The authors suggest that the initial difference in abundance of the [3Fe-4S] species between MS and EPR in Figure 8C is due to the [3Fe-4S]^0^ species, which is EPR inactive (Subsection “The first step of 4Fe-4S] RirA cluster conversion is the O_2_-independent reversible loss of Fe^2+^ to form [3Fe-4S]”). Can it be ruled out that the lack of EPR signal is simply due to the regulator converting from the [4Fe-4S]^2+^ state to the [3Fe-4S] state, in an oxygen dependent manner? This reaction would also explain the observation of the resolved buildup in the first couple of minutes of the [3Fe-4S]^1+^ species by EPR.

We are not sure that we follow this point. The reviewers ask if we can ‘rule out that the lack of EPR signal is simply due to the regulator converting from the [4Fe-4S]^2+^ state to the [3Fe-4S] state, in an oxygen dependent manner’. They don’t specify that charge associated with the [3Fe-4S] cluster, but because they’re referring to an EPR silent form, we presume they mean [3Fe-4S]^0^? If so, then we are not sure why we would conclude that this is O_2_-dependent, particularly when we observe the same rate constants in the presence and absence of O_2_. We think that the simplest interpretation of the data is that [3Fe-4S]^0^ is generated as the first step, and this is oxidised to the [3Fe-4S]^1+^ state when O_2_ is present. That [3Fe-4S]^0^ is the first intermediate under anaerobic conditions is supported by the *K*_d_ determination measurements now included in the manuscript.

Does Fe dissociate from the [4Fe-4S] also in the absence of chelator? An explanation for the difference with MS could be that due to the ionization conditions of ESI-MS, temporal resolution of the initiation of the reaction is not possible. Without this resolution, the kinetic analysis that results from the speciation diagrams in Figure 4 and Figure 7 could be difficult to interpret.

We’re not sure which difference the reviewers are referring to. We measured the cluster conversion/degradation under identical conditions by ESI-MS and absorbance, observing the same rate constants for the initial reaction in both. Therefore, it is clear that the ESI-MS is faithfully reporting on the kinetics of cluster breakdown. Iron continuously dissociates from and re-associates with the RirA cluster in the absence of chelator. Re-association is affected in the presence of an iron-binding competitor.

As displayed in Figure 2A, the ionization conditions of ESI-MS lead to many Fe-bound species at t = 0.

Yes, we acknowledge this issue. We think that this is due to the variable loss of some [4Fe-4S] clusters during the exchange of the RirA sample into ammonium acetate buffer for ESI-MS measurements, resulting in some cluster breakdown species. To investigate this further, we measured changes in the ESI-MS spectra recorded at 5 min compared to that after 37 min of RirA under anaerobic conditions in the absence of an iron chelator, see Figure 2—figure supplement 1. We observed that the [3Fe-2S] and [3Fe-3S] peaks reduced in intensity, while the [2Fe] and [2Fe-2S] increased, indicating that these part-degraded clusters continue to degrade. There is no apparent drop off in [4Fe-4S] and only minor increases in apo-forms, indicating that the [4Fe-4S] form that survived the buffer exchange is stable in the absence of either an iron chelator or O_2_.

Can the authors exclude that small amounts of oxygen at the injection port of the MS or in the dry gas may lead to the observed cluster decomposition at t = 0?

The answer above is also relevant here – if O_2_ were an issue, we would continually observe the breakdown of clusters. After sitting in the injection syringe for half an hour, cluster breakdown products had decayed further. This is consistent with our previous measurements of O_2_-mediated FNR cluster degradation by ESI-MS (Crack et al., 2017). In that study we used the same methodology to exclude O_2_ as employed here. We were able to measure the [4Fe-4S] FNR species for 2 hours in the absence of O_2_ without decomposition, demonstrating that O_2_ is not a significant factor in the injection port (which is flushed with deoxygenated buffer prior to measurements).

Comparing Figure 2A and Figure 6A, there is sample variation before the addition of chelator, which may be the result of small amounts of oxygen. O^2^ levels can be much better controlled in EPR sample preparation. Therefore overall, we feel it cannot be definitely concluded that O^2^ is reacting with the [3Fe-4S]^0^ species and not the [4Fe-4S] cluster.

There is some variation in the extent of cluster breakdown products from one experiment to another (as further emphasised in the figure above). We think this arises from some variation in the extent to which the [4Fe-4S] is partially degraded during the buffer exchange, and not due to reaction with O_2_. Furthermore, if [4Fe-4S] were reacting directly with O_2_ we would see an O_2_ dependence of the reaction. We don’t – we observed the same rate for the initial reaction (by ESI-MS and absorbance) in the absence and presence of O_2_ (as mentioned in the manuscript, and this was also reported in Pellicer Martinez et al., 2017). The variation in the starting spectrum in terms of relative intensities does not affect the relative intensity behaviour of species as a function of time, as these are entirely reproducible.

[Editors' note: the author responses to the re-review follow.]

Essential revisions:Abstract: The phrase "…for which Kd = ~3 μM, consistent with this equilibrium sensing 'free' iron in the cell cytoplasm." sounds awkward and is not connected to the first part of the sentence. Further, it is not really explained in the Abstract how the authors envision iron and oxygen regulation. Please re-write and make these points clear.

The Abstract is limited to 150 words and we so cannot expand explanations. However, we have re-written the Abstract within the word limit to try to make clearer the key points on iron- and O_2_-sensing.

Figure 4 and Figure 7: It is not fully clear to us why an apo species appears in Figure 4E (at least 70%) and 7F (plateau at 90%) but the A(382nm) does not decay substantially. As far as we understand, the cluster in MS analysis is gone after 30 min (all MS-detectable species are close to zero), but the colour at 382 nm remains. Please clarify and explain in text. Minor point: It may be didactically better to present Figure 4 and Figure 7 in the very same order.

This is a good point. Significant A382 nm intensity remains at t=30 min, even though ESI-MS demonstrates that no [4Fe-4S] remains at this point and the major species detected are apo-forms of RirA. A382 nm arises from a range of contributing transitions, including from [2Fe-2S] and [2Fe] and other forms of Fe/S. Firstly, both [2Fe-2S] and [2Fe] are present in the 30 min ESI-MS spectrum under anaerobic conditions, and there is some [2Fe] under aerobic conditions (and we note that the ionization efficiency of species vary and so the relative abundances do not necessarily directly reflect absolute solution concentrations).

Secondly, and more importantly, Fe/S that were previously part of the Fe-S cluster remain, either bound to the protein, weakly attached, or in solution/suspension as Fe-S species, or as iron acetate (the red-brown colour of Fe(III) acetate was commonly used to quantitate Fe(III) in solution). All of these contribute to A382 nm but only Fe/S species tightly bound to the protein (which thus survives ionization) are detected by ESI-MS. The nature of the residual species may be different under anaerobic/aerobic conditions, and the greater percentage decrease in A382 nm observed in the presence of O_2_ correlates with the enhanced level of sulfide oxidation observed under aerobic conditions (Figure 7), which would limit the availability of sulfide for complexation with released iron. Evidence for low molecular weight species, generated during cluster conversion, that contribute to A382 nm was previously reported (ref 28, supplementary data). In that study, a significant decrease in absorbance following passage of chelator-exposed [4Fe-4S] RirA down a gel filtration column was observed. The uncertainty of what A382 nm is due to essentially reflects the need to find new methods, such as we have applied here, that can determine which protein-bound FeS species are actually present.

We have added sentences to the legends of Figure 4 and Figure 7 to acknowledge/account for this point. We have also edited Figure 7 so that data are presented in the same order as in Figure 4.

It is also unclear how the presence of the initially present cluster breakdown products affects the kinetic data and their modelling. The authors state that "The variation in the starting spectrum in terms of relative intensities does not affect the relative intensity behaviour of species as a function of time, as these are entirely reproducible." This is hard to comprehend – how can what is there at the start of a reaction not affect subsequent reactions? Perhaps the answer lies in what "the relative intensity behaviour" reflects and how it is used to arrive at the quantitative data, but this complication needs to be made clearer in the manuscript.

Firstly, the variation in the starting spectrum will indeed lead to variation in the relative intensities of intermediate species – these are represented by the error bars in the plots of Figure 4 and Figure 7. The behaviour of the relative intensities (in terms of increases/decreases) was found to be entirely reproducible, allowing rate constants to be estimated from the modelling. The starting intensities due to intermediates (which we have shown arise from some cluster breakdown during buffer exchange) were dealt with in the modelling by allowing these to be offset from zero at t = 0 (clearly no mechanism exists that can account for intermediates at t = 0!). We have clarified these issues in the manuscript (Materials and methods section).

Figure 5: The first equilibrium is shown for [4Fe-4S]^2+^ going to a [3Fe-4S]^1+^. Wouldn't it be more appropriate to show the [3Fe-4S]^0^ species as the product of this first equilibrium, which then is converted to [3Fe-4S]^1+^ in the presence of oxygen?

Yes, we see the point here and agree – we have edited the figure.

Figure 1—figure supplement 1: The figure shows four highlighted Cys residues but only three are expected to be cluster coordinating. The homology model predicts that Cys17 is not close enough for cluster binding. This should be stated/clarified. It could also be stated that in the NsrR structure, Asp8 is the fourth coordinating residue. This Asp is not conserved in RirA.

Thank you for pointing this out – we have removed highlighting of Cys17 and added the points mentioned to the legend.

One of the major previous concerns was the relation of the reported experiments to physiological context. In response, as requested by the reviewers, a dissociation constant for Fe(II) has been determined, even though – as the authors explain – they were concerned that this might not be possible due to the fragility of the [3Fe-4S]^0^ cluster formed in the initial "sensing" reaction. Surprisingly, the equilibrium titration worked nonetheless, with only ca. 1 Fe transferred to Mag-fura-2. An explanation why the subsequent cluster decay did, therefore, not happen (as the liberated Fe would also be picked up by the chelator) within the time frame of the experiment is missing. The time frame is mentioned but not specified. Was the titration done on a single solution, where subsequent additions of Mag-fura-2 were added to the same solution? Or does each point correspond to a different solution?

Thank you for the comment and we can see that we weren’t sufficiently clear in our description of the *K*_d_ experiments. We initially established that the first Fe^2+^, but not more than this, was available for chelation essentially immediately (i.e. complete before we could measure an absorbance spectrum), with cluster conversion occurring over a much longer time period (several hours, as previously reported). This was an important finding because it meant that the suggested *K*_d_ experiments were indeed feasible. The titrations were done with single solutions of RirA, to which increasing concentrations of mag-fura-2 were added. Because Fe^2+^ chelation occurred rapidly, there was no requirement for prolonged incubation of samples between additions, and each titration was completed in ~60 mins, enabling us to measure the equilibrium distribution of the single available Fe^2+^ between RirA and the chelator before any significant cluster degradation. We have added further details of the *K*_d_ measurements to the Materials and methods section.

Further major concerns were related to erroneous kinetic constants – the ensuing corrections have addressed these concerns to a significant degree. This includes questions around the effects of oxygen.However, in the discussion regarding the identity of the various adducts, the authors mention that O_2_ can be generated in the ESI-MS source, and that this could be the origin of the adducts with oxygen. It is important to acknowledge this, and to clarify whether or not this phenomenon affects the results.

We have addressed this in the section of the manuscript entitled ‘The effect of O_2_ on [4Fe-4S] RirA cluster conversion by mass spectrometry’, where we now further acknowledge that some O_2_ may be generated during the ESI-MS experiment. We conclude, though, that the significant differences between the anaerobic and aerobic datasets indicate that O_2_ generated during the ESI-MS experiment is minor. Indeed, as we previously mentioned in our first response to reviewers comments, the fact that we can readily detect O_2_-sensitive [4Fe-4S] FNR using the same methodology is consistent with relatively minor amounts of O_2_. Again, the differences observed between the anaerobic and aerobic datasets indicate that any O_2_ generated does not significantly affect the results, and we note that the O adducts observed are confined to apo-RirA.

The question how the (relatively slow) kinetics observed in the current work relate to the processes occurring in vivo has been addressed in the response letter, but not necessarily in the manuscript. The conclusions mention continuous dissociation and association, but considering that kinetics are at the heart of the paper, it would be nice to have a comment on timescales here.

We have included in the conclusion section a paragraph exploring the timescales of sensing processes, along the lines of our response to the reviewers’ initial comments.

Finally, although the authors do a beautiful job exploring the disassembly of the 4Fe-4S cluster in RirA free in solution, they do not consider the impact that the bound DNA might have, in any way. If this is a true sensing mechanism then DNA binding by the repressing 4Fe-4S complex may well impact both the rate of dissociation of the "fourth Fe" as well as its equilibrium affinity (or not). Although we realize that such an investigation is well beyond the scope of this manuscript, simply ignoring this essential aspect of this system is less than satisfactory. We suggest the authors work a bit of this into their conclusions. Is it known whether the metallated RirA-DNA complex can be studied by native ESI-MS?

This is a good point and one that we are actively working on. Our previous studies of NsrR and FNR bound to DNA led to the conclusion that their reaction mechanisms with nitric oxide and O_2_, respectively, are not affected by DNA-binding (Crack et al., 2016 and Crack et al., 2014), but this may not be the case for iron/O_2_-sensing by RirA and that is something we plan to investigate. And, yes, we are exploring the potential to study regulator-DNA complexes using ESI-MS. We do actually raise some of the important remaining questions about RirA in the final paragraph of the conclusions, including aspects of DNA-binding. But we did not specifically raise the point about effects on *K*_d_ or cluster conversion mechanism, and so we have now included these outstanding future challenges in the final paragraph.